# Structural dynamics of the E6AP/UBE3A-E6-p53 enzyme-substrate complex

Carolin Sailer[1], Fabian Offensperger[1], Alexandra Julier[1], Kai-Michael Kammer[1], Ryan Walker-Gray[2], Matthew G. Gold [2], Martin Scheffner[1] & Florian Stengel [1]

Deregulation of the ubiquitin ligase E6AP is causally linked to the development of human disease, including cervical cancer. In complex with the E6 oncoprotein of human papillomaviruses, E6AP targets the tumor suppressor p53 for degradation, thereby contributing to carcinogenesis. Moreover, E6 acts as a potent activator of E6AP by a yet unknown mechanism. However, structural information explaining how the E6AP-E6-p53 enzyme-substrate complex is assembled, and how E6 stimulates E6AP, is largely missing. Here, we develop and apply different crosslinking mass spectrometry-based approaches to study the E6AP-E6-p53 interplay. We show that binding of E6 induces conformational rearrangements in E6AP, thereby positioning E6 and p53 in the immediate vicinity of the catalytic center of E6AP. Our data provide structural and functional insights into the dynamics of the full-length E6AP-E6-p53 enzyme-substrate complex, demonstrating how E6 can stimulate the ubiquitin ligase activity of E6AP while facilitating ubiquitin transfer from E6AP onto p53.

[1] Department of Biology, University of Konstanz, Universitätsstrasse 10, 78457 Konstanz, Germany. [2] Department of Neuroscience, Physiology & Pharmacology, University College London, Gower Street, London WC1E 6BT, UK. These authors contributed equally: Carolin Sailer, Fabian Offensperger. Correspondence and requests for materials should be addressed to M.S. (email: martin.scheffner@uni-konstanz.de) or to F.S. (email: florian.stengel@uni-konstanz.de)

In eukaryotes, covalent modification of proteins by ubiquitin (ubiquitylation) plays a pivotal role in the regulation of many key cellular processes, including cell cycle, DNA metabolism (e.g., DNA repair), and various signal transduction pathways[1–3]. The specificity of the ubiquitin–conjugation system is mainly ensured by E3 ubiquitin ligases, which mediate the recognition of target proteins. Based on the presence of distinct domains and their mode of action, E3 ligases have been grouped into three families, RING/RING-like E3s, RING-between-RING (RBR) E3s, and HECT E3s[4,5]. In a simplified view, all E3s have at least two binding sites, one for substrate proteins and one for cognate E2 ubiquitin-conjugating enzymes. However, while RBR E3s and HECT E3s form a covalent intermediate with ubiquitin and transfer it to their targets, RING/RING-like E3s function mainly as adaptors facilitating the direct transfer of ubiquitin from the E2 to the substrate.

Deregulation of ubiquitylation—at the substrate level or at the level of the conjugation machinery—has been causally involved in the development of human disease, including cancer, neurological disorders, and viral infections[1,6–9]. A prominent example is provided by E6AP, the founding member of the HECT (homologous to E6AP C terminus) E3 family[10,11]. E6AP was originally identified as a cellular interaction partner of the E6 oncoprotein of so-called high-risk human papillomaviruses (HPVs) such as HPV-16, which cause anogenital cancers, most notably cancer of the uterine cervix[12,13]. In complex with E6, E6AP targets the tumor suppressor p53 and other proteins—which in the absence of E6 are not targeted by E6AP—for ubiquitylation and degradation by the 26S proteasome, thereby contributing to HPV-induced cervical carcinogenesis[14–16]. E6AP is encoded by the UBE3A gene, which is located on chromosome 15q11-13. In 1997, it was shown that genetic alterations of the UBE3A gene, resulting in loss of E6AP expression or in the expression of E6AP variants with compromised E3 activity, are the cause of the Angelman syndrome (AS), a neurodevelopmental disorder[17–20]. However, although several potential substrate proteins of E6AP have been reported, including HHR23A and HHR23B, AIB1, PML, alpha-Synuclein, Ring1B, and Arc[21–27], the pathophysiologic relevance of these interactions remains mostly unclear. More recently, it was reported that amplification of the chromosomal region containing the UBE3A gene is found in individuals with Dup15q syndrome, an autism spectrum disorder[28,29]. Although final proof that amplification of the UBE3A gene underlies the Dup15q syndrome is missing, experiments in mice and fruit flies support the notion that increased E6AP levels result in autistic phenotypes[30,31].

The findings that alteration of the substrate spectrum (cervical cancer), loss of E3 function (AS), and increased E3 function (Dup15q) of E6AP contribute to the development of distinct disorders indicate that expression and/or E3 activity of E6AP have to be tightly controlled. Indeed, in certain brain areas the paternal allele is silenced by a UBE3A antisense transcript (i.e. in these areas, E6AP is mainly expressed from the maternal allele and genetic alterations of the maternal allele are responsible for AS development)[32], and recent evidence indicates that the phosphorylation status of T485 (numbering according to isoform 1 of human E6AP[33]) affects E6AP activity[34]. In addition, there is strong evidence that the E3 activity of E6AP can be activated by interacting proteins. HERC2, a giant HECT E3, binds to a region within the N-terminal 200 amino acid residues of E6AP and somehow stimulates the E3 activity of E6AP[35]. Mutations in the HERC2 gene, which result in increased degradation of HERC2 and, thus, in decreased expression levels, have been etiologically associated with an AS-like syndrome pointing to the importance of the HERC2-E6AP interaction[36]. Similarly, the E6

oncoprotein not only alters the substrate spectrum of E6AP, but also acts as a potent activator of E6AP by a yet unknown mechanism[37].

The data obtained with HERC2 and the E6 oncoprotein[35,37] suggest that E6AP exists in at least two inter-converting conformational states of higher and lower activity. However, while the structure of the catalytic HECT domain was solved almost two decades ago[38], there is little structural information concerning full-length E6AP or for the N-terminal part of E6AP, with the exception of the AZUL domain[39] and the E6 binding region[40].

Similarly, the crystal structure of E6 bound to an E6AP-derived peptide of 12 amino acids containing the LQELL motif and to the DNA binding domain of p53 was recently solved[41]. While revealing the role of the E6-binding region of E6AP in assembling the E6–p53 interaction, the structure did not provide any insight into how full-length E6AP is oriented towards E6 and p53 in the E6AP–E6–p53 complex and how the E3 activity of E6AP becomes stimulated by interaction with E6.

Yet, this information is essential for a molecular understanding of how interaction partners regulate E6AP activity. Thus, in this study, we developed and applied different approaches in structural mass spectrometry (MS), in particular, chemical cross-linking coupled to MS (XL-MS) to study the effect of the E6 oncoprotein on E6AP conformation. The general approach of XL-MS is to introduce covalent bonds between proximal functional groups of proteins or protein complexes by crosslinking reagents. The actual crosslinking sites are subsequently identified by MS and reflect the spatial proximity of regions within a given protein (intralinks) or of subunits in a protein complex (interlinks) (for recent reviews, see refs. [42,43]). Thereby, XL-MS provides a wealth of information on the connectivity, interaction, and relative orientation of regions within a protein and of subunits within a complex, and also contains spatial information in itself, though at a relatively low resolution.

Using qualitative and quantitative XL-MS approaches, we show that binding of the E6 oncoprotein induces conformational changes within full-length E6AP. In addition, we provide evidence that E6 does not only contact the originally identified E6 binding site of E6AP, but also comes into close proximity to the catalytic HECT domain and positions the HECT domain next to its substrate p53. Taken together, our data provide molecular and structural insights into how the E6 oncoprotein stimulates the E3 activity of E6AP as well as into the full-length E6AP–E6–p53 enzyme–substrate complex.

## Results

**Binding of E6 induces conformational rearrangement of E6AP.** Even though reasonable amounts of highly purified E6AP can be generated, attempts to obtain high-resolution structures of full-length E6AP, by ourselves and other groups, have so far been unsuccessful for unknown reasons. Therefore, XL-MS was applied as an alternative method to obtain insight into protein structure and in particular into structural rearrangements induced by an interacting partner[42–44]. We applied isotopically labeled disuccinimidyl suberate (DSS) to crosslink highly purified E6AP (Supplementary Figure 1) and determined the crosslinking pattern within full-length E6AP (for a general overview of the susceptibility of lysine residues to crosslinking in this study, see Supplementary Figure 2). We identified 145 unique lysine–lysine contact sites (uxIDs) within full-length E6AP, corresponding to 396 unique crosslinks that were identified in total over three biological replicates (Supplementary Data 1). Figure 1a shows the overall distribution of high-confidence crosslinks that were reproducibly and consistently identified between different

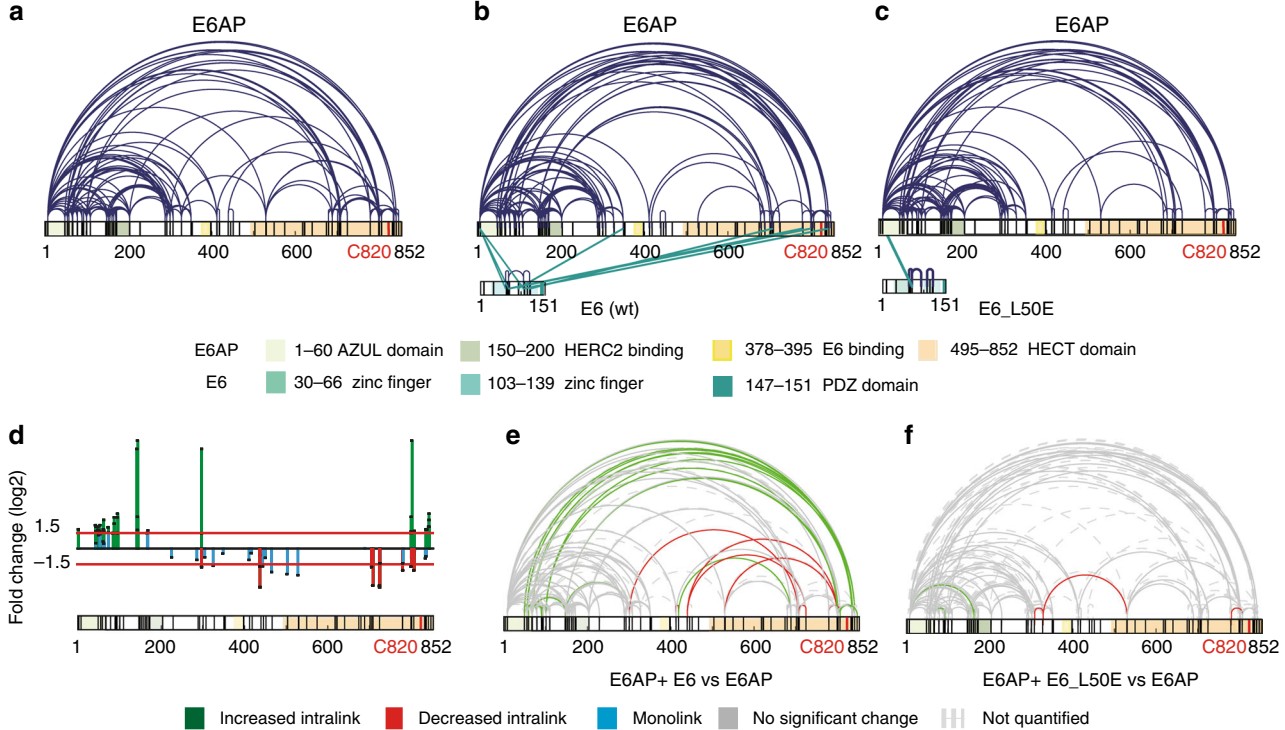

**Fig. 1** Binding of E6 induces a conformational rearrangement of E6AP. Pattern of intralink distribution within E6AP as determined by XL-MS **a** in the absence of the HPV E6 oncoprotein, **b** with wild-type (wt) E6, and **c** with the E6_L50E mutant, which does not bind to E6AP. Intralinks are shown in dark blue and interlinks in dark green; lysine residues are shown in black. The catalytic cysteine residue of E6AP at position 820 is indicated in red. The AZUL domain, HERC2 and E6 binding sites, and the HECT domain are indicated in pastel green, sulfur yellow and sand yellow, respectively. Regions in E6 representing zinc finger motifs are colored in pastel turquoise and the PDZ binding domain is colored in mint turquoise. **d** Changes in crosslink abundance for each unique crosslinking site with E6AP upon binding of wild-type E6 were used to calibrate and normalize the quantitative XL-MS (q-XL-MS) data. Changes are expressed as fold change ($\log_2$ ratio of abundance of E6AP in the presence of E6 versus abundance of E6AP alone). The horizontal red line indicates the significance threshold (fold change: log2 ratio $\leq \pm 1.5$). Changes in monolinks are shown in blue and significantly changed intralinks within E6AP are shown in green (relative increase upon binding of E6) and red (relative decrease upon binding of E6), respectively. The $p$ values for each quantified link are indicated. **e** Applying q-XL-MS to recombinantly expressed E6AP in the presence or absence of wild-type E6 identifies numerous high-confidence crosslinks. Only crosslinks that could be reproducibly quantified from the pool of identified high-confidence crosslinks in both samples (with and without E6) ($n = 3$, each sample analyzed additionally as technical duplicate) and consistently over 3 different biological replicates are shown (violation = 0; $p$ value $\leq 0.01$ (two-sided $t$-test)). Depicted in green are crosslinks that were found to be significantly upregulated upon binding of E6, while downregulated links are shown in red (defined as a log2 change of $\geq \pm 1.5$). Crosslinks with no significant change are depicted in gray while links that could not be reliably quantified are show with a dashed line in light gray. **f** Quantification of the change in abundance of identified intralinks within E6AP, when incubated with the binding-deficient E6_L50E mutant vs. E6AP alone, reveals no significant up- or downregulated links. For monolinks, see Supplementary Figure 5

biological replicates. Crosslinks were identified over the entire length of E6AP, encompassing all known functional regions, i.e., the AZUL and HECT domains and the HERC2 and E6 binding sites. We next repeated the experiment in the presence of the HPV-16 E6 oncoprotein (E6; note that a GST-E6 fusion protein was employed), identifying 146 unique uxIDs, corresponding to 408 unique crosslinks that were identified in total over three biological replicates (Supplementary Data 2). Remarkably, in addition to the expected interlinks between E6AP and E6, the presence of E6 had a noticeable effect on the crosslinking pattern within E6AP itself, leading to diminished crosslinking within the central region of the protein (i.e., the region of E6AP between the AZUL and the HECT domain) (Fig. 1b). To corroborate that changes in the crosslinking pattern were caused by interaction between E6AP and E6, we repeated the experiment using the L50E mutant of E6 (E6_L50E; a GST fusion protein was employed). E6_L50E binds only weakly to E6AP, if at all[40], and consequently does not stimulate the E3 activity of E6AP, as demonstrated by its inability to rescue E6AP autoubiquitylation

in the presence of the hydrophobic patch mutant UbLIA (Supplementary Figure 3)[37]. In the presence of E6_L50E, we identified 131 unique lysine–lysine contact sites, which correspond to 325 unique crosslinks identified in total over three biological replicates (Supplementary Data 3; for a direct comparison of all crosslinked peptides under the various conditions see Supplementary Data 4). Importantly, interlinks between E6AP and the E6 mutant were almost completely absent, supporting previous findings that the ability of the mutant to bind E6AP is impaired[40] and demonstrating the veracity of our approach. Moreover, unlike wild-type E6, E6_L50E had little effect on the crosslinking pattern within E6AP itself (Fig. 1c). Taken together, the distinct crosslinking patterns obtained suggest that E6AP undergoes a structural rearrangement upon binding of E6.

The above findings are in line with our recent experiments showing that the E6 oncoprotein acts as a potent activator of the E3 function of E6AP[37] in addition to affecting the substrate spectrum of E6AP. In fact, our initial crosslinking measurements are consistent with E6 stabilizing a high-activity conformation of

E6AP. To dissect the structural basis of this reorganization in more detail, we next employed differential or quantitative XL-MS (qXL-MS), whereby identified crosslinks are quantified via their MS1 or MS2 intensities[43]. This technique is a recent innovation[43,45] that is proving to be valuable for analyzing protein conformations under different conditions[44]. In order to normalize the data and also to identify the subset of significantly changed crosslinks, we plotted the quantitative change in the abundance of monolinks within E6AP in the absence and presence of E6 (Fig. 1d), reasoning that monolink formation is dictated primarily by accessibility of a lysine residue and its local environment (e.g., surrounding amino acid composition, secondary structure etc.) and therefore only on rare occasions, if at all, influenced by conformational dynamics of the protein. Doing this, we identified the vast majority of changes in monolink abundances fluctuating within a range of log2 ratio ≤ ±1.5. Thus, in this study, only changes that showed at least a change of log2 ratio ≥ ±1.5 and a p-value of ≤ 0.01 using a two-sided t-test were considered significant changes in crosslink abundance.

Applying the above criteria to the set of identified crosslinks allowed us to probe the relative changes in the crosslinking patterns of E6AP in the absence and presence of E6 and to quantify a number of 129 unique linking sites (106 crosslinks and 23 monolinks) over different biological states and replicates (Supplementary Data 5). Figure 1e shows all 106 crosslinks that were reproducibly and consistently quantified (violation = 0; p value ≤ 0.01 (two-sided t-test); Supplementary Figure 4). Shown in green are crosslinks that were found to be increased or upregulated upon binding of E6, while decreased or downregulated links are shown in red (defined as a log2 change of ≥ ±1.5).

The majority of crosslinks did not substantially change in abundance (depicted in gray, Fig. 1e), indicating that large areas of E6AP are unaffected by the presence of E6. However, crosslinking was markedly altered within two prominent features that clearly indicate conformational dynamics within E6AP. First, multiple crosslinks within the HECT binding domain as well as emanating from it toward the E6 binding region decreased in the presence of E6 (red, Fig. 1d, e). However, more striking were a collection of crosslinks between the N- and C-terminal regions of E6AP that were significantly upregulated upon binding of E6, in some cases exceeding a 100-fold increase in intensity (green, Fig. 1d, e; Supplementary Data 5, 13, 14). To confirm that these changes resulted from *binding* of E6 to E6AP, rather than its mere presence, we performed equivalent q-XL-MS experiments with the E6_L50E mutant. As expected, no significant up- or downregulated links were detected (Fig. 1f, Supplementary Figure 5, Supplementary Data 6, 15, 16). Taken together, our q-XL-MS data therefore indicates that upon binding, E6 induces a conformational rearrangement of E6AP that brings the N- and C-terminal regions into closer proximity.

**SILAC-XL-MS reveals weak interactions between E6AP N-termini.** Determination of the crystal structure of the isolated HECT domain of E6AP revealed that the HECT domain can assume a homo-trimeric conformation[38]. While solution binding experiments and mutational analysis indicated that trimer formation was an artifact of crystal packing[38], more recent kinetic and modeling experiments[46,47] suggest that to display E3 ligase activity, E6AP may have to form homo-trimers. A possible explanation for these seemingly contradictory results is that oligomeric E6AP-E6AP interactions are transient and too fleeting to be detected by conventional solution binding assays. In principle, low affinity protein–protein interactions can be stabilized by suitable crosslinkers[48]. However, conventional XL-MS cannot

readily distinguish between intra-subunit crosslinks and inter-subunit crosslinks between individual protomers within a homo-oligomer (with the exception of sequence identical peptides that are bijective within the particular protein sequence). While it has been shown that homodimeric complexes can in principle be reconstituted from mixtures of $^{14}N/^{15}N$-labeled subunits[49–52], this approach is not readily extensible to larger complexes. To address this shortcoming, we combined XL-MS with stable iso-tope labeling with amino acids in cell culture (SILAC)[53].

We first tested and validated our SILAC-XL-MS workflow using the known homodimer Glutathione S-transferase (GST) from *Schistosoma japonicum*. We expressed GST in an *E. coli* strain that is auxotrophic for lysine and arginine[54] using minimal media supplemented with either light lysine/arginine or heavy isotope-labeled lysine (D4; K + 4.025108 Da)/arginine (13C6 & 15N4: R + 10.00826 Da). "Heavy" and "light" GST were purified separately. Subsequently, the resulting protein preparations were crosslinked and analyzed using an adapted version of the xQuest software suite (Methods). We then compared the pattern of links after crosslinking either light GST alone (Supplementary Data 7), or a mixture of light and heavy GST (Supplementary Data 8). As expected, "heavy" peptides were only identified in samples containing heavy GST (Supplementary Figure 6A). Crosslinks between "light" and "heavy" peptides clustered to lysines close to the dimer interface, with the highest scoring peptide bridging lysines K10 and K124, which align at the dimer border (Supplementary Figure 6B). Having validated our approach for analyzing oligomeric contacts by XL-MS and SILAC with a known dimer, we next analyzed E6AP using the same approach. Control experiments confirmed that SILAC labeling of E6AP had no influence on its E3 activity (Supplementary Figure 6C). "Light" and "heavy" E6AP preparations were mixed in a 1:1 ratio. As E6AP oligomers are not stable under the conditions used for purification (Supplementary Figure 6D and Methods), this should result in the formation of mixed oligomers consisting of "light" and "heavy" E6AP molecules, assuming E6AP oligomers exist.

Homomeric interlinks within E6AP detected by SILAC-XL-MS are shown in Fig. 2 in the absence (Fig. 2a) and presence of E6 (Fig. 2b). Here, we identified 109 unique lysine–lysine contact sites (uxIDs) within full-length E6AP in the absence of E6, corresponding to 379 unique crosslinks (containing 204 light–light, 168 heavy–heavy, and 7 light–heavy links) that were identified in total over three biological replicates (Supplementary Data 9). In the presence of E6, we identified 108 unique lysine–lysine contact sites (uxIDs) within full-length E6AP, corresponding to 383 unique crosslinks (containing 204

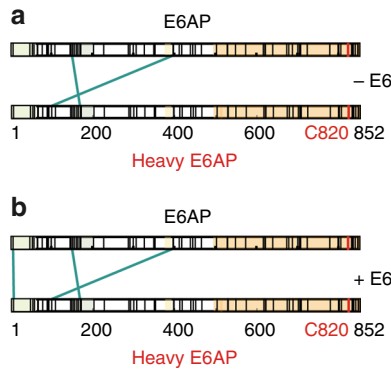

**Fig. 2** SILAC-XL-MS reveals weak interactions between E6AP N-termini. Identified interlinks between two E6AP protomers (i.e., heavy and light version of E6AP). Shown are the interlinks in the absence (**a**) and presence of E6 (**b**)

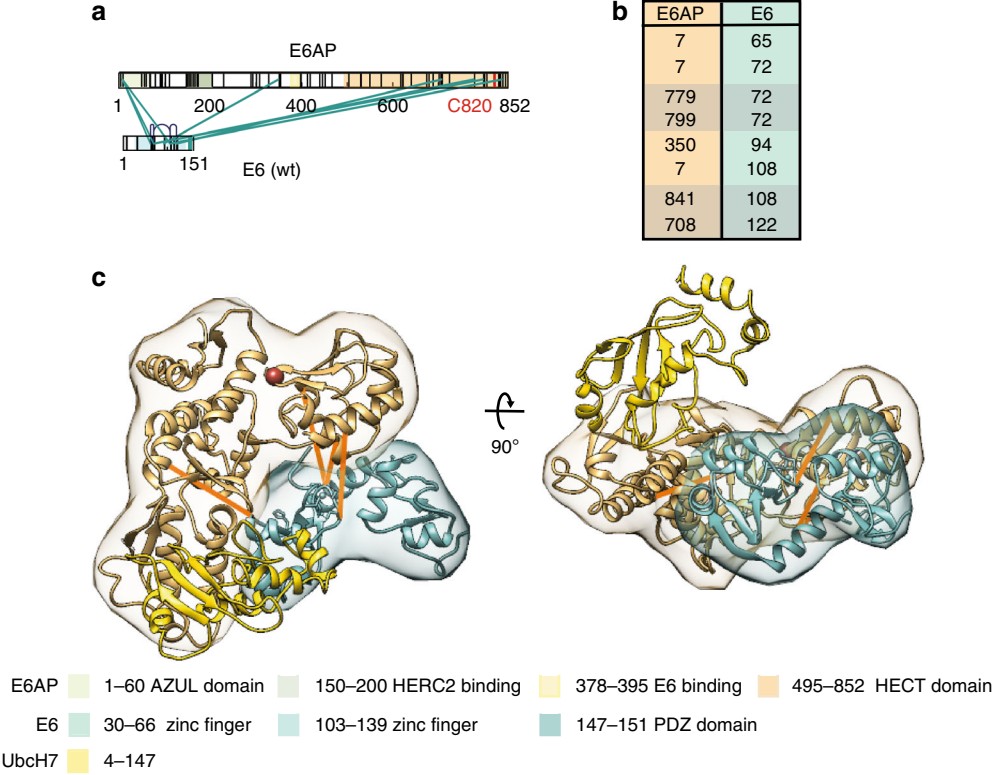

**Fig. 3** E6 contacts the HECT domain of E6AP. **a** Shown are the identified interlinks between E6AP and E6. Lysine residues and regions of known functions within E6AP and E6 are color-coded as in Fig. 1. **b** List with identified interlinks between E6AP and E6. Numbering is according to human E6AP isoform 1[33]. Links which can be mapped to existing PDB structures (4XR8; 1C4Z) are highlighted. **c** Structural model of the binding interface between the HECT domain of E6AP and E6. The localization densities of the HECT domain of E6AP (residues 495–846) and E6 (residues 1–151) are shown in sand yellow and light green, respectively, with a single representative ribbon structure embedded. For clarity, UbcH7, a cognate E2 of E6AP, was also added and is shown in yellow (PDB: 1C4Z). Detected interlinks between E6 and E6AP that map to PDB structures 4XR8 and 1C4Z, respectively, are highlighted in orange. The catalytic cysteine on position C820 in the HECT domain of E6AP is shown as a red ball. The model is shown from the top (left) and from a side view (right)

light–light, 172 heavy–heavy, and 7 light–heavy links) (Supplementary Data 10). Comparison of the two patterns of light–heavy interlinking indicates that E6 does not significantly alter E6AP oligomerization, as suggested previously[46]. The identified homodimeric links are located in the N-terminal part of E6AP, i.e., the N-termini interact with each other in the establishment of the E6AP homo(di)mer, even though the relatively low number of identified interlinks cautions us to make an unambiguous assignment in terms of binding sites within the E6AP homo(di)mer. However, it is interesting to note that none of the detected interlinks is located within the C-terminal half of E6AP comprising the HECT domain, where the oligomerization site has been previously proposed to be located[46,47]. In summary, while the absence of crosslinks does not unambiguously prove that there is no interaction, our SILAC-XL-MS data suggest that E6AP can engage in homo-oligomeric interactions, but these are likely not mediated through the HECT domain, and that oligomerization is not initiated or intensified by E6.

**E6 contacts the HECT domain of E6AP.** We previously demonstrated that XL-MS can be used to predict and identify protein–protein interaction sites with high precision[55]. We therefore had a closer look at the identified crosslinks between E6AP and E6 (Fig. 3a, b). Up to now, a major part of E6AP has resisted structural analysis. This region spans residues 60-500 (numbering according to human E6AP isoform 1[33]) and is predicted to be mostly folded, except for two small intrinsically unfolded stretches comprising residues 100–150 and 370–400, which contain or are adjacent to the interaction regions for the E3

ligase HERC2 and E6, respectively[35,56]. Nonetheless, the majority of identified interlinks between E6AP and E6 are located within the three regions of E6AP that could be resolved at high resolution (Fig. 3b, Supplementary Data 2). One of these links, E6AP_K350 – E6_K94, confirms the known E6 binding site, as it links E6 with the lysine that is closest to the small part of the central region of E6AP (residues 380–394) encompassing the helical LQELL motif, which is bound to E6 in the crystal structure[40,41,56]. Some identified crosslinks fall within the very N-terminal region (residues 1–64) which contains a zinc-binding module termed AZUL domain and whose structure has been solved by NMR[39]; however, as we identify crosslinks to K7 of E6AP in all experiments, this interaction should be regarded with a certain amount of reservation, as it might be caused by the very flexible and reactive nature of this particular lysine residue.

Remarkably, most of the identified interlinks fall within the C-terminal region (residues 495–852) of E6AP that comprises the HECT domain, the structure of which has been solved by X-ray crystallography[38]. Altogether, we were able to detect 4 unique uxID crosslinking sites between E6AP and E6 with a total of 10 unique crosslinks (and a total of 127 unique crosslinks including intralinks) within the HECT domain of E6AP. Intriguingly, all of the interlinks are located around or very near the catalytically active cysteine in E6AP at position C820[11]. These crosslinks connect the HECT domain of E6AP directly with the two homologous zinc-binding domains in E6, which themselves adopt an α–β fold connected by a linker helix[40,41].

The richness of the crosslinking data allowed us to model this previously undescribed binding site using Bayesian crosslinking

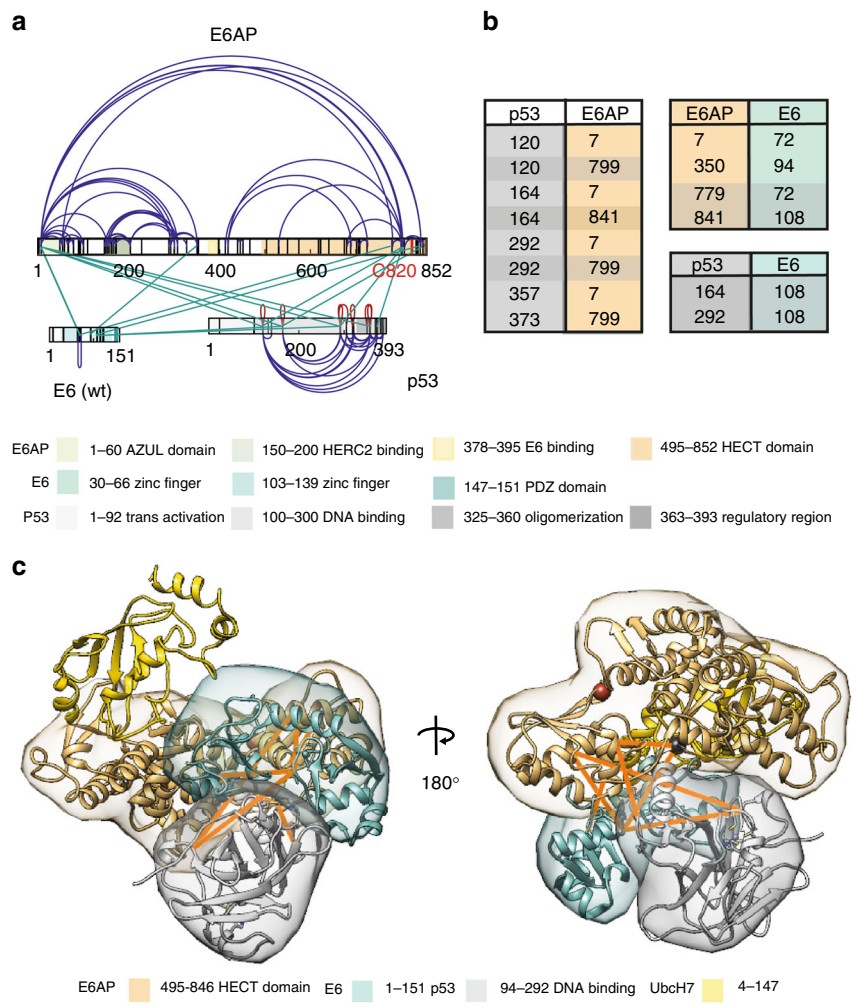

**Fig. 4** Distinctive binding sites within the E6AP–E6–p53 complex. **a** Shown are the identified interlinks between E6AP, E6, and p53. Lysine residues and regions of known functions within E6AP and E6 are color-coded as in Fig. 1. Functional domains in p53—transactivation domain, DNA binding domain, oligomerization domain and regulatory region—are shown in signal white, gray white, telegrey, and white aluminum, respectively. Crosslinks between sequence identical peptides that are bijective within the particular protein sequence are shown as red loops. **b** List with identified interlinks between p53, E6AP, and E6. Numbering is according to human E6AP isoform 1[33]. Links which can be mapped to existing PDB structures (4XR8; 1C4Z) are highlighted. **c** Structural model of the binding interface between the HECT domain of E6AP, E6, and p53. The localization densities of the HECT domain of E6AP (residues 495–846), E6 (residues 1–151), and p53 (residues 94–292) are shown in sand yellow, light green, and telegrey, respectively, with a single representative ribbon structure embedded. For clarity, UbcH7, a cognate E2 of E6AP was also added and is shown in yellow (PDB: 1C4Z). Detected interlinks between E6, p53, and E6AP that map to PDB structures 4XR8 and 1C4Z, respectively, are highlighted in orange. The catalytic cysteine at position C820 in the HECT domain of E6AP is shown as a red ball. Residue K292 of p53 (most C-terminal lysine residue in the model) is shown as a gray ball. The model is shown from a side view (left), using the exact same view as in Fig. 3 right panel, and rotated around its axis (right)

guided integrative structural modeling[57,58]. Using the crosslinking dataset (Supplementary Data 2) and the high-resolution structures of the HECT domain of E6AP (PDB: 1C4Z) and of E6 (PDB: 4XR8) as an input for our modeling approach (see Methods), we derived an unbiased, highly reproducible and robust model for the E6AP–E6 binding interface, where the sampling runs converged onto one main cluster of structural solutions (Supplementary Figure 7). The average precision for the E6 main cluster was ~6 Å, while E6AP was used as reference to align to, using a root-mean-square deviation (rmsd) cutoff for clustering of 5 Å (Supplementary Figure 7).

In this model, E6 is firmly placed next to the catalytic center of the HECT domain and adjacent to the known E2 binding site (for clarity, the cognate E2 of E6AP, UbcH7, was added[38]) (Fig. 3c). Our crosslinking data together with the crosslinking-based modeling therefore identify a E6AP–E6 binding site that is in direct vicinity to the catalytic center of the HECT domain and does not overlap with the known E2 binding site[38]. This indicates

that the HECT domain can simultaneously interact with E6 and cognate E2s, thereby ensuring efficient ubiquitylation of substrate proteins.

**Distinctive binding sites within the E6AP–E6–p53 complex.** Besides the interaction with E6 and cognate E2s, it can be postulated that the HECT domain of E6AP also comes into close proximity to substrate proteins for subsequent ubiquitylation. The tumor suppressor p53 is the best characterized substrate of the E6–E6AP complex[14–16]. Thus, we included purified recombinant p53 (Supplementary Figure 9) to assemble the E6AP–E6–p53 enzyme–substrate complex. Following crosslinking with DSS, we were able to identify 105 unique lysine–lysine contact sites, corresponding to 305 unique crosslinks that were identified in total over three biological replicates within and between full-length E6AP, E6, and p53 (Supplementary Data 11).

We identify several links that firmly connect the DNA binding region of p53 with the HECT domain of E6AP and also with the N-terminal zinc finger in E6. There is also an additional crosslink between the HECT domain of E6AP and the C-terminal 30 amino acids of p53 (E6AP_K799 - p53_K373) (Fig. 4a, b; Supplementary Data 11). Concerning the interaction of E6 with E6AP, the links not only contained the one close to the helical LQELL E6 binding motif but also the majority of the crosslinks between E6 and the HECT domain identified above in the experiments to Fig. 3a, b (Fig. 4a). In fact, looking at the picture that emerges from the entirety of identified E6AP–E6 interlinks, the crosslinking pattern between E6AP and E6 in the presence of p53 almost exactly corresponds to the one in the absence of p53, indicating that the identified binding pattern between E6AP and E6 is also relevant during the formation of the enzyme–substrate complex (Fig. 4a, b; Supplementary Data 11). Moreover, the same two lysine residues that are located within the HECT domain of E6AP and which make up half of the unique crosslinking sites between E6AP and E6 are also involved in crosslinks formed between p53 and E6AP. This indicates that within the E6AP–E6–p53 enzyme–substrate complex, E6 and p53 are positioned next to each other and in close proximity to the catalytic center of the HECT domain.

Applying crosslinking guided integrative modeling corroborates these findings. Using the crosslinking dataset from the E6AP–E6–p53 complex (Supplementary Data 11) and including E6 and p53 into a single rigid body[41] in order to enforce the established interface between the two proteins leads again to a highly reproducible and robust model, where sampling runs converged onto one main cluster (Supplementary Figure 7). For the E6AP–E6–p53 complex, the average precision for the main cluster was ~7 Å for p53 and ~5 Å for E6, while E6AP was again used as reference to align to with an rmsd cutoff for clustering of 5 Å (Supplementary Figure 7). Comparing this model to our model on the E6AP–E6 alone dataset and using the same view as in Fig. 3c (right panel) shows that the E6AP–E6 binding interface between E6AP and E6 is very similar (Fig. 4c, left panel). In our model of the E6AP–E6–p53 interface, E6 is firmly placed next to the catalytic center of the HECT domain and adjacent to the E2 binding site (for clarity, UbcH7 was also added). Here, p53 is located adjacent to E6 and the catalytic center of E6AP (Fig. 4c, left panel). Rotating the model around its axis highlights the distance between the potential ubiquitylation sites within the DNA binding domain of p53[59] and the catalytic cysteine residue of E6AP (Fig. 4c, right panel).

In conclusion, our crosslinking data together with the crosslinking-based modeling confirms that the previously undescribed E6AP–E6 binding site is also formed in the presence of a substrate (i.e., p53) and strongly indicates that both E6 and the substrate p53 are located in direct vicinity to the catalytic center of E6AP within the enzyme–substrate complex.

## Discussion

This study presents a set of experiments where different approaches in structural mass spectrometry were applied in order to obtain insight into how (i) the E6 oncoprotein of cancer-associated HPVs stimulates the E3 ligase activity of E6AP, and (ii) E6AP, E6, and the tumor suppressor p53 are structurally and functionally arranged within the E6AP–E6–p53 enzyme–substrate complex. It also highlights an innovative approach to study structural aspects and dynamics of protein–protein interactions within E3 ligase-substrate complexes, in particular when using full-length proteins.

In a first set of experiments, crosslinking of E6AP in the absence and presence of the HPV-16 E6 oncoprotein revealed distinctive crosslink patterns, which suggested that E6AP undergoes a structural rearrangement upon binding of E6. This finding is in line with recent data, where we showed that in addition to affecting the substrate spectrum of E6AP, the E6 oncoprotein is a potent activator of E6AP[37]. While this indicated that E6AP can adopt high and low activity conformations, structural evidence for this hypothesis was lacking prior to this study. Using q-XL-MS and our developed data analysis workflow that allows to normalize the data and exclude potential influences on the crosslinking behavior not caused by conformational dynamics of the protein complex itself, we obtained evidence that binding of the E6 oncoprotein induces a conformational rearrangement within E6AP that brings the N- and C-terminal regions into closer proximity and/or stabilizes interactions between these regions. This finding strongly supports the conclusion derived from biochemical analyses that the N-terminal region of E6AP plays an important role in E6AP-mediated ubiquitin chain formation[37].

The isolated HECT domain of E6AP can homo-trimerize in the context of a crystal lattice[38] but the functional relevance of this oligomerization is not clear. However, recent kinetic and modeling experiments[46,47] suggest that in order to function as an E3 ligase, E6AP may have to form homo-trimers. To resolve these ambiguities, and to test for possible low affinity interactions between different E6AP molecules that are not readily detected by conventional means, we used XL-MS in combination with SILAC. Data obtained by this approach, which we termed SILAC-XL-MS, indicate that E6AP is capable of engaging in homo-oligomeric contacts. However, these interactions are not mediated via the HECT domain and do not appear to be initiated or intensified by the E6 oncoprotein, as suggested previously[46,47]. While it cannot be excluded that the GST E6 fusion protein used in this study was, at least in part, oligomeric, this also clearly indicates that the oligomerization status of E6 does not affect the oligomerization behavior of E6AP. To obtain further insight into the potential oligomerization behavior of E6AP, it will be important to compare the crosslinking pattern of the isolated N-terminal region of E6AP (i.e., a truncation mutant of E6AP devoid of the HECT domain), the isolated HECT domain, and full-length E6AP. We also noticed that the number of uxIDs are lower in experiments, where SILAC was used in combination with crosslinking. This may be explained by a direct effect of SILAC on the efficiency of crosslink identification. However, as the number of overall identified unique crosslinks are roughly similar in SILAC and non-SILAC crosslinking experiments, this effect may alternatively be explained by the relative smaller amount of peptides that was loaded on the column (i.e., the same total peptide amount loaded onto the LC MS/MS column should translate to half the amount of light E6AP peptides being present in the SILAC vs the non-SILAC sample).

Next, we used XL-MS and integrative modeling to probe protein–protein contact sites between E6AP and the E6 oncoprotein. In doing so, we not only confirmed the known interaction site between E6AP and E6[40,41,56], but more importantly, identified an additional E6AP–E6 binding interface that is in direct vicinity to the catalytic cysteine residue of the HECT domain and does not overlap with the known E2 binding site[38]. This could in principle be explained by two E6AP binding sites on one E6 molecule or by binding of two E6 molecules to one E6AP molecule. Although we cannot distinguish between these possibilities directly, the crosslinks identified clearly show that two different regions of E6 are involved in the interaction with the known E6 binding site and with the HECT domain. It is therefore likely that one E6 molecule interacts with one E6AP molecule. This suggests the exciting possibility that upon binding, the E6 oncoprotein not only induces a conformational change in E6AP

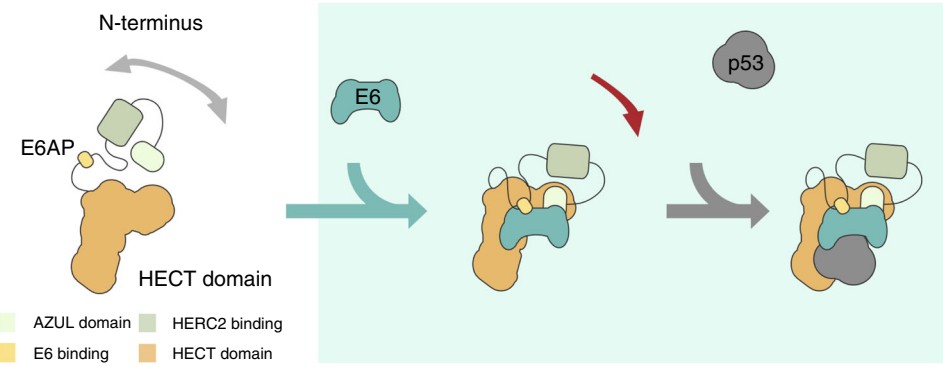

**Fig. 5** Model of E6-mediated ubiquitylation of p53. Model of E6-mediated ubiquitylation of p53 in the E6AP–E6–p53 enzyme–substrate complex. Upon binding, E6 induces a conformational rearrangement in E6AP that brings the AZUL domain closer to the HECT domain. This leads to the concomitant positioning of E6 and the substrate p53 into the direct vicinity of the catalytic center of E6AP within the enzyme–substrate complex. There, E6 now functions as an adaptor facilitating the direct transfer of ubiquitin from the HECT domain of E6AP to p53

but in addition, may function as a cofactor in E6AP-catalyzed isopeptide bond formation.

In a last set of experiments, we used XL-MS and integrative modeling to probe protein–protein contact sites between E6AP, the E6 oncoprotein, and their substrate p53. Here, we identified several interlinks between the HECT domain of E6AP and the DNA binding region as well as the C-terminal 30 amino acids of p53. Since the same lysine residues within the HECT domain are part of crosslinks that are formed between E6AP and E6 and crosslinks that are formed between p53 and E6AP, our data strongly suggest that both E6 and p53 are simultaneously in contact with the HECT domain and in close proximity to its catalytic center within the E6AP–E6–p53 enzyme–substrate complex. As MS is an ensemble technology, it is fully conceivable that the HECT domain within the E6AP–E6–p53 enzyme–substrate complex in the one case forms a crosslink with the neighboring E6 oncoprotein and in another with the adjacent p53 molecule. In this context, it is important to note that p53 forms a tetramer and this is presumably also the form of p53 that was present within the E6AP–E6–p53 complex investigated in this study, as indicated by the multiple identified crosslinks (red loops) between sequence identical peptides in p53 (Fig. 4a). This is also in line with previous XL-MS studies of p53 alone, which identified intralinks and interlinks between different p53 molecules that were all located within the oligomerization domain and the C-terminal 30 amino acids of p53[52]. In our study, we could not only identify several of those links, but detected a significantly larger amount of crosslinks that connect the DNA binding region with the C-terminal 30 amino acids of p53. We therefore cannot discriminate by XL-MS for a given crosslink between E6AP and p53, with which of the p53 subunits of the proposed tetramer it interacts. This (p53 tetramer) may explain why in our model, the distance between the catalytic center of E6AP and the nearest potential ubiquitylation site (K292) in the modeled p53 (i.e. DNA binding domain) is still ~12 Å (assuming 5 Å and 3 Å flexibility for a lysine or cysteine residue, respectively), a distance too large for a nucleophilic attack (i.e. transfer of ubiquitin from the cysteine residue of E6AP to the lysine residue of p53). Moreover, there is an additional crosslink between the HECT domain of E6AP and the C-terminal 30 amino acids of p53 (E6AP_K799 - p53_K373). As this region of p53 was not included in our modeling runs, as it is predicted to be disordered, it is possible that lysine residues within this region of p53 are even closer to the catalytic center of E6AP than K292. Alternatively, it can be envisioned that the conformational rearrangement within E6AP that is induced by E6 binding (vide supra) may also result in a

slight conformational rearrangement within the HECT domain itself, which would bring the catalytic center into the required distance for the transfer of ubiquitin onto p53. Furthermore, a similar rearrangement of the HECT domain may occur, when ubiquitin is bound to the catalytic cysteine residue via a thioester bond.

To address the latter possibility, it would be necessary to generate an E6AP population that is loaded with ubiquitin but in such a way that ubiquitin cannot be transferred from E6AP onto p53. Although this may in principle be possible (e.g., by exchange of the catalytic cysteine residue by a lysine residue[60]), it is a rather challenging task to generate amounts sufficient for XL-MS. Nonetheless, our crosslinking data together with the crosslinking-based modeling confirms that the previously undescribed E6AP–E6 binding site is also formed in the presence of a substrate (i.e., p53) and strongly indicates that both E6 and p53 are located in direct vicinity to the catalytic center of E6AP within the enzyme–substrate complex.

Our structural understanding of the role of E6AP in E6–E6AP-p53 complex formation and in E6-mediated ubiquitylation of p53 was so far limited to a 12-amino-acid peptide derived from E6AP[41]. It was shown that E6, once bound to the LQELL motif of E6AP, exposes on its surface a large cleft that interacts with the p53 DNA binding domain (residues 94–292). Yet, while revealing the role of the E6-binding region of E6AP in assembling the E6–p53 interaction, the structure did not provide any insight into how full-length E6AP is oriented toward E6 and p53 in the E6AP–E6–p53 complex and how the E3 activity of E6AP becomes stimulated by interaction with E6. Our approach did not only reveal distinctive binding sites within the E6AP–E6–p53 enzyme–substrate complex, but also put us into the position to propose a model, by which the E3 activity of E6AP is activated in order to transfer ubiquitin onto its substrates (Fig. 5). Upon binding, E6 induces a conformational rearrangement in E6AP that brings the AZUL domain closer to the HECT domain, such allowing the concomitant positioning of E6 and the substrate p53 into the direct vicinity of the catalytic center of E6AP. There, E6 functions as an adaptor, and possibly as an enzymatic cofactor (see above), facilitating the direct transfer of ubiquitin to p53.

In summary, by structural mass spectrometry we attained structural and functional insights into the dynamics of the full-length E6AP–E6–p53 enzyme–substrate complex, paving the way for a general understanding of how E6AP activity is controlled and how it can be manipulated in the treatment of E6AP-associated clinical pictures.

## Methods

**Protein expression and purification.** His-tagged human E1 enzyme (UBA1) was expressed in *E. coli* BL21 DE3 and purified via Ni-NTA affinity chromatography[61]. The hydrophobic patch ubiquitin mutant UbLIA was expressed in *E. coli* BL21 DE3 and purified by size exclusion chromatography[35]. His-tagged UbcH5b was expressed in *E. coli* BL21 DE3 and purified via Ni-NTA affinity chromatography[37]. Fractions containing UbcH5b were pooled, dialyzed against 25 mM Tris-HCl pH 7.5, 50 mM NaCl, 1 mM DTT, and stored at −80 °C.

His-tagged E6AP was expressed in *E. coli* Rosetta DE3 at 20 °C overnight. Cell pellets derived from 1 L bacterial culture were resuspended in 30 mL of 25 mM Tris-HCl pH 7.5, 50 mM NaCl, 0.1% Triton, 1 mM DTT, 1 μg/mL aprotinin and leupeptin, and 100 μM Pefabloc. After sonication and centrifugation (15,000 × g, 4 °C, 15 min), the supernatant was loaded onto a Ni-NTA affinity chromatography column (HisTrapTM FF crude, 5 mL column), washed with 8 column volumes of buffer A (25 mM Tris-HCl pH 7.5, 50 mM NaCl, 1 mM DTT) followed by a gradient of 20 column volumes to 100% buffer B (25 mM Tris-HCl, 50 mM NaCl, 500 mM imidazole, pH 7.5). Fractions containing E6AP were pooled and subjected to a second purification step by anion exchange chromatography (HiTrapTM Q HP, 1 mL column), using a gradient from 0 to 50% buffer B in 20 column volumes (buffer A: 25 mM Tris-HCl pH 7.5, 50 mM NaCl, 1 mM DTT; buffer B: 25 mM Tris-HCl pH 7.5, 1 M NaCl, 1 mM DTT). 4 mL Amicon filter devices with a cutoff of 10 kDa were used for buffer exchange to 25 mM Tris-HCl pH 7.5, 50 mM NaCl, 1 mM DTT and to concentrate the sample. Note that E6AP eluted at ~320–340 mM NaCl, a condition that was also used for size exclusion chromatography (Supplementary Figure 6).

For SILAC, an *E. coli* strain auxotrophic for arginine and lysine[54] was used to incorporate 4,4,5,5-deuterated lysine (K + 4.025108 Da) and 13C$_6$/15N$^4$-labeled arginine (R + 10.00826 Da) (Cambridge Isotope Laboratories). Expression was performed overnight at 20 °C in M9 minimal media (7.52 g/L Na$_2$HPO$_4$, 3 g/L KH$_2$PO$_4$, 0.5 g/L NaCl, 0.5 g/L NH$_4$Cl) containing 0.4% glucose, 1 mM MgSO$_4$, 0.3 mM CaCl$_2$, 1 μg/L biotin, 1 μg/L thiamine, 134 μM EDTA, 31 μM FeCl$_3$, 6.2 μM ZnCl$_2$, 0.76 μM CuCl$_2$, 0.42 μM CoCl$_2$, 1.62 μM H$_3$BO$_3$, 81 nM MnCl$_2$. For SILAC-XL-MS validation experiments, we expressed isotope-labeled GST using the vector pGEX6P1. Bacterial pellets were lysed using sonication in buffer containing 30 mM HEPES pH 7.5, 500 mM NaCl, 2 mM DTT, 0.5 mM EDTA, and 1 mM benzamidine. GST was purified by batch affinity to glutathione sepharose 4B (GE Healthcare) and size exclusion using a Superdex 200 column in 20 mM Na Hepes pH 7.5, 200 mM NaCl. Unlabeled GST was purified in the same way following expression in LB media. Following purification, both unlabeled and isotope-labeled GST dimers were dissociated by supplementing the buffer with 6 M urea to induce dimer dissociation[62]. Dissociated labeled and isotope-labeled monomers were then recombined, and urea was removed by serial dilution to enable formation of mixed unlabeled-isotope-labeled dimers. To confirm that treatment with urea did not affect the conformation of dimeric GST, we performed XL-MS also with untreated light GST. The crosslinking patterns for untreated GST (Supplementary Data 12), and GST subjected to urea unfolding/refolding (Supplementary Data 7), are highly similar. We also performed control experiments with light GST alone to test the accuracy of isotope-labeled peptide assignment.

Purification of isotope-labeled E6AP was performed as described above for unlabeled E6AP. As oligomeric E6AP interactions are transient under the conditions used for purification (Supplementary Figure 6D), there was no need to dissociate light–light and heavy–heavy E6AP dimers prior to cross-linking. GST-fusion proteins of wild-type HPV-16 E6[12] and the E6_L50E mutant[40] were expressed in *E.coli* BL21-CodonPlus-RIL at 37 °C for 4 h. Pellets derived from 1 L bacterial culture were resuspended in 30 mL PBS, 0.1% Triton, 1 mM DTT and sonicated, and the lysate cleared by centrifugation (16,000 × g, 4 °C, 15 min). The supernatant was incubated with 150 μL glutathione-sepharose 4B (GE Healthcare) at 4 °C for 1 h. Beads were spun down and washed three times with 1 mL PBS. GST-E6 fusion proteins were eluted four times with 400 μL 25 mM Tris-HCl pH 8.0, 50 mM NaCl, 25 mM gluthatione. 4 mL amicon filter devices with a cutoff of 10 kDa were used for buffer exchange to 25 mM Tris pH 7.5, 50 mM NaCl, 1 mM DTT and to concentrate the sample.

His-lipoyl domain-tagged p53 was expressed in *E. coli* BL21 DE3 at 24 °C overnight[63]. Cells were lysed in 50 mM phosphate buffer pH 8.0, 300 mM NaCl, 0.01% NP-40, 1 μg/mL aprotinin and leupeptin, 100 μM Pefabloc, 10 mM DTT. After sonication the lysate was centrifuged for 20 min at 15,000 × g and the supernatant was loaded onto a Ni-NTA column. His-tagged p53 was eluted with a gradient of 25 column volumes from buffer A (50 mM phosphate buffer pH 8, 150 mM NaCl, 0.01% NP-40, 2 mM DTT, 25 mM imidazole) to 100% buffer B (50 mM phosphate buffer pH 8, 150 mM NaCl, 0.01% NP-40, 2 mM DTT, 1 M imidazole) and dialyzed overnight (50 mM phosphate buffer pH 7.2, 150 mM NaCl, 0.01% NP-40, 2 mM DTT) in the presence of thrombin (Sigma-Aldrich). Cleaved p53 was separated from the His-tagged lipoyl domain using a heparin column. p53 was eluted from the heparin column with a gradient over 25 column volumes from buffer A (50 mM phosphate buffer pH 7.2, 150 mM NaCl, 0.01% NP-40, 2 mM DTT) to 100% buffer B (50 mM phosphate buffer pH 7.2, 1 M NaCl, 0.01% NP-40, 2 mM DTT).

**Ubiquitylation assays.** For in vitro ubiquitylation, 150 ng E1, 75 ng UbcH5b, 750 ng E6AP, and 2 μg ubiquitin (bovine, Sigma-Aldrich) or UbLIA were incubated in 25 mM Tris-HCl pH 7.5, 50 mM NaCl, 1 mM DTT, 2 mM MgCl$_2$, 2 mM ATP at 30 °C in a total volume of 30 μL. Reactions were stopped after 90 min or the indicated time by addition of 7.5 μL 5× sample buffer (312.5 mM Tris-HCl pH 6.8, 10% SDS, 500 mM DTT, 0.001% bromphenol blue) and electrophoresed in 12% SDS-polyacrylamide gels followed by Coomassie blue staining.

**Crosslinking coupled to mass spectrometry.** Complexes were crosslinked and measured essentially as described[64]. In short, ~100 μg of E6AP were crosslinked by addition of H12/D12 DSS (Creative Molecules) at a ratio of 1.5 nmol/1 μg protein for 30 min at 37 °C shaking at 650 rpm in a Thermomixer (Eppendorf). In order to crosslink E6AP in complex, wild-type E6 or the E6_L50E mutant (and p53) were added in a molar ratio of 1:1(:1) and incubated on ice for 30 min prior to cross-linking under the same conditions. Proteins were crosslinked directly after purification without freezing. After quenching by addition of ammonium bicarbonate to a final concentration of 50 mM, samples were reduced, alkylated, and digested with trypsin. Digested peptides were separated from the solution and retained by a solid phase extraction system (SepPak, Waters), and then separated by size exclusion chromatography prior to liquid chromatography (LC)-MS/MS analysis on an Orbitrap Fusion Tribrid mass spectrometer (Thermo Scientific). Data were searched using xQuest in ion-tag mode with a precursor mass tolerance of 10 ppm. For matching of fragment ions, tolerances of 0.2 Da for common ions and 0.3 Da for crosslink ions were applied. Crosslinked samples were prepared in biological triplicates (i.e., separately expressed and purified batches of proteins) for all investigated samples, and each of these was measured with technical duplicates. Crosslinks were only considered during structural analysis, if they were identified in at least two of three biological replicates with deltaS < 0.95 and at least one Id score ≥25. A list of all identified links can be found in Supplementary Data 1–16).

**Quantitative crosslinking coupled to mass spectrometry.** For quantitative crosslinking coupled to mass spectrometry (q-XL-MS) analysis, the chromatographic peaks of identified crosslinks between E6AP in the absence and presence of E6 or E6_L50E were integrated and summed up over different peak groups (taking different charge states and different unique crosslinked peptides for one unique crosslinking site into account) for quantification by xTract[45]. Amounts of potential crosslinks were normalized prior to MS by measuring peptide bond absorption at 215 nm for each fraction. Only high-confidence crosslinks that were identified consistently over different biological and technical replicates in a peak group (xTract settings violations was set to 0) were selected for further quantitative analysis. Changes in crosslinking abundance are expressed as log$_2$ ratio (e.g., abundance state 1, E6AP with E6 was quantified versus abundance state 2, E6AP alone). The *p* value using a two-sided *t*-test indicates the regression between the two conditions.

The question of how to assess the significance of a measured quantitative change in a crosslink in terms of relevance for a potential conformational change on the protein level has to the best of our knowledge not been addressed so far in the crosslinking field; contrary to the sheer reproducibility or statistical validity of a measured crosslinking change, for which different statistical evaluation criteria have been proposed. Thus, in order to normalize the data and to identify the subset of significantly changed crosslinks, the quantitative distribution of monolinks within E6AP in the absence and presence of E6 or E6_L50E was plotted (see Fig. 1d and Supplementary Figure 5), reasoning that monolink formation is dictated primarily by accessibility and its local environment (e.g., surrounding amino acid composition, secondary structure etc.) and therefore only on rare occasions influenced by conformational dynamics. Doing this, we identified the vast majority of changes in monolink abundances fluctuating within a range of log2 ratio ≤ ±1.5. Thus, in this study, only changes that showed at least a change of log2 ratio ≥ ±1.5 and a *p*-value of ≤0.01 for the E6 or ≤0.05 for the E6_L50E dataset were considered significant changes in abundances and are shown in green and red in the 2D visualizations (Fig. 1d–f; Supplementary Figure 5), respectively. All other changes were considered insignificant and are shown in gray.

**SILAC-based crosslinking coupled to mass spectrometry.** In order to distinguish between intralinks within a protomer and interlinks between protomers of a homomeric interaction, we incorporated the use of stable heavy isotopes via SILAC (stable isotope labeling with amino acids in cell culture)[53] into our XL-MS workflow, which we termed SILAC-based crosslinking coupled to mass spectrometry or SILAC-XL-MS. For SILAC-XL-MS, unlabeled GST or E6AP and GST or E6AP expressed in the presence of stable heavy isotopes are mixed in order to form hetero-oligomers of unlabeled and isotope-labeled proteins, which are crosslinked and after MS measurement analyzed by a modified version of our xQuest analysis platform. In short, GST or E6AP were expressed in an *E. coli* strain that is auxotrophic for lysine and arginine[54] in the absence or presence of stable heavy isotopes (lysine (D4; K + 4.025108 Da) and arginine (13C6 & 15N4: R + 10.00826 Da)), as described above (Protein expression and purification). Upon expression and purification, the two GST/E6AP preparations were mixed in a 1:1 ratio (prior to mixing, GST preparations were treated with 6 M urea to induce dissociation of light–light and heavy–heavy dimers, see above) and the samples were subsequently crosslinked and measured on an Orbitrap Fusion Tribrid mass spectrometer. Subsequently, the data was searched by an adapted version of xQuest. XQuest was

adapted in a way that it contained two additional artificial amino acids (B = K + 4.025108 Da and U = R + 10.00826 Da), an additional crosslinking site (B), and an artificial enzyme that cuts at K|R|B|U to allow for an in silico tryptic digestion.

**Visualization of crosslinks**. Visualization of intra- and interlinks was performed by xiNET[65] using additional in-house scripts for the analysis and representation of quantitative crosslink information. For the static datasets all links that passed the evaluation criteria—Id scores > 25 in at least 1 biological replicate, a deltaS value < 0.95 and detection in at least two out of the three independent biological replicates—are shown. For the differential (quantitative) datasets in Fig. 1d–f and Supplementary Figure 5, only links that were consistently quantified in all biological datasets (violation 0) were shown.

**Integrative structural modeling**. We used the Integrative Modeling Platform (IMP)[57] for modeling the structure of the HECT domain of E6AP and E6 with and without p53. The approach using crosslinks as restraints in a Bayesian scoring scheme is described in detail in ref. [58]. Accordingly, there are four main steps: (1) gathering of data, (2) representation of subunits and translation of the crosslinking data and the prior knowledge into a Bayesian scoring function, (3) configurational sampling to produce an ensemble of models that minimize the Bayesian scoring function, and (4) analysis of the ensemble. IMP allows for coarse-grained modeling, i.e. the inclusion of different resolution levels into a model. Different resolution levels are represented by accordingly sized beads (spheres) in a modeling run.

In the Bayesian scoring scheme models are ranked according to their likelihood and prior probability. The score is the negative logarithm of their product. The likelihood contains the crosslinking data, while the priors contain information about sequence connectivity in protein chains as well as excluded volume between all pairs of beads. The likelihood is defined through a forward model, which quantifies the probability of the formation of a crosslink given the distance between two residues in the model, as well as a noise model, which weighs the deviation between observed crosslinks and the forward model.

For our modeling runs, we employed the known crystal structures of the HECT domain of E6AP (PDB ID: 1c4z) and E6/p53 (PDB ID: 4xr8). In the modeling runs, the following residues were represented by beads (one residue per bead) and constrained into rigid bodies: for E6AP residues 495–846 of chain A, for E6 residues 1–151 of chain F and for p53 residues 94–292 of chain C. Residues 144–151 of E6 in chain C are not represented in the crystal structure and were grouped into a single flexible bead. E6 and p53 were included into a single rigid body in order to enforce the established interface between the two proteins. The crosslink input databases for the modeling included all links (until ld-20) which were found in at least two out the three biological replicates. From each replicate the highest scoring link was chosen.

The actual models were computed by Replica Exchange Gibbs sampling, based on Metropolis Monte Carlo sampling[66]. The Monte Carlo movements included random translation and rotation of rigid bodies with a maximum of 10 Å and 1 radian, respectively.

The sampling was run on 16 replicas producing 15,000 models each, with temperatures ranging from 1.0 to 2.5 (technical units). The 25% best scoring models were saved, leading to 60,000 saved models overall. These replicas were run in three independent sampling runs with random initial configurations to assess convergence and accounting for 180,000 models overall.

The models from all three sampling runs were pooled and the 500 best scoring models from each sampling run were clustered using the root-mean-square deviation (rmsd) of E6 and p53 (when p53 was included in the sampling). The HECT domain of E6AP was used as reference to align to, as its position was fixed during the sampling. The rmsd cutoff for clustering was 5 Å. The cluster center was defined as the cluster member whose rmsd with respect to the other members is minimal. It was used to represent the atomistic coordinates of the system. The overall precision of each protein was calculated as the mean root-mean-square fluctuation with respect to the cluster center. Furthermore, to represent the variance of the solutions, the superposed structures of each cluster were converted into a localization density[67,68].

For each setup (i.e., with and without p53), the sampling runs converged onto one main cluster. To assess convergence, all replicates were also clustered independently, resulting in the same main clusters as the pooled clustering. The average precision for the E6 main cluster in the modeling runs not containing p53 was ~5 Å. In the run containing p53, it was ~8 Å for p53 and ~5 Å for E6.

To assess robustness, the modeling procedure described above was repeated while randomly removing 15% of the crosslinks (jackknifing). Additionally, we also repeated the modeling using only crosslinks with ld-scores greater than 30. In both cases the sampling runs converged onto the same clusters with precisions within 1 Å of the original runs, as can also be seen in the rmsd matrices produced by these runs (Supplementary Figure 7). Additional modeling runs, in which all interlinks between E6AP and E6 for a specific uxID were purposely removed, as well as an independent modeling approach using HADDOCK confirm the overall robustness and reproducibility of our modeling results (Supplementary Figure 8).

**Code availability**. We have compiled a repository of all the required changes to the xQuest software. The git repository containing all modified xQuest files and instructions on how to install them may be found on github.com (https://github.com/stengellab/silac_xl_ms).

## Data availability

All data generated or analyzed during this study are included in this published article (and its supplementary information files). The MS data (raw files, xQuest and xTract files) have been deposited to the ProteomeXchange Consortium via the PRIDE[69] partner repository with the dataset identifier PXD010002. Both the structural model of the binding interface between the HECT domain of E6AP and E6 (PDBDEV_00000022) and the structural model of the binding interface between the HECT domain of E6AP, E6, and p53 (PDBDEV_00000023), including localization densities, have been deposited in PDB-dev.

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

## Acknowledgements

We thank Thomas Walzthoeni for help with the adaption of the xQuest software suite for SILAC-XL-MS, Greta Marie Assman, Kathy Su, Kay Diederichs for help using HADDOCK and Riccardo Pellarin for help with the integrative structural modeling. This work was supported by the Konstanz Research School Chemical Biology (KoRS-CB). F.O. acknowledges funding from the Zukunftskolleg of the University of Konstanz. M.G.G. is a Wellcome Trust and Royal Society Sir Henry Dale fellow (104194/Z/14/Z) and receives support from the BBSRC (BB/N015274/1). F.S. is funded by the German Science Foundation Emmy Noether Programme (STE 2517/1-1). F.S. and M.S. are grateful for support from the DFG Collaborative Research Centre (SFB) 969.

## Author contributions

C.S., F.O., M.S., and F.S. conceived the study and experimental approach; F.O. and A.J. cloned, expressed and purified E6AP, E6, and p53. F.O. carried out ubiquitylation assays; C.S., K.M.K., F.O., and A.J prepared samples for XL-MS experiments. C.S., K.M.K., F.O., R.W.G., M.G.G., and F.S implemented and carried out SILAC XL-MS experiments. C.S. performed XL-MS and q-XL-MS experiments; K.M.K. carried out the integrative modeling; C.S., F.O., M.S., and F.S. analyzed the data, and M.S. and F.S. wrote the paper with input from all authors.
