## [Peer Review File · Nature Communications]

Reviewers' comments:

Reviewer #1 (Remarks to the Author):

In HPV-induced cancers, the E6 HPV oncoprotein (~150 residues) primes the E6AP HECT ubiquitin ligase (~850 residues) to poly-ubiquitylate tumor suppressor p53 (~390 residues). It is known that the binding to E6 strongly activates the ubiquitin ligase activity of E6AP, not only towards p53 but also to other substrates of E6AP, including E6AP itself. A prior crystallographic structure has shown that full-length E6 protein interacts with a helical "LxxLL" motif derived from the central region of E6AP to recruit the "core" central domain of p53 (Martinez-Zapien et al., Nature 2016). However, high-resolution structural data concerning full-length E6AP alone or complexed to full-length E6 and/or full-length p53 are still missing.

In this manuscript, Sailer et al have applied state-of-the-art cross-linking mass spectrometry (XL-MS) to map 1) the intramolecular contacts between different regions of E6AP in the absence of E6 and p53; 2) the intramolecular contacts between different regions of E6AP in the presence of E6 and of E6 and p53; 3) the intermolecular contacts between different regions of two different molecules of E6AP in the absence of E6 and p53; 4) the contacts established by full-length E6AP with E6 and p53 within pre-formed E6-E6AP and E6-E6AP-p53 complexes. They have produced highly reproducible data which allow them to map the different contacts listed above, and extract exciting information about the conformational changes occurring to E6AP upon E6 binding, which may explain the resulting "hyper activation" of the E6AP enzymatic activity. Furthermore, based on the cross-linking patterns they compute distance-driven models of the relative spatial arrangements of the different domains of E6, E6AP, p53 and E2 ubiquitinase ligase UbcH7 in a quaternary complex of these different proteins.

The experimental data are very convincing, and as far as I can judge (not being an expert in mass spectrometry approaches) extremely carefully performed and analyzed. The conclusions are very sound and convincing and will represent a very important contribution to our understanding of the fascinating mechanism of E6-E6AP mediated degradation of p53. My comments here will be only related to my field of expertise (biochemistry and structural biology).

1) It would be useful to represent (supplemental figure the sequences of the three proteins used in the study enlightening all lysine residues, with a different color for the lysines which have reacted and the ones which did not react.

2) It would be very useful to include in the figures additional tables in the same format as the tables shown in Fig 4B and Fig5B, to represent E6AP to E6AP contacts in the absence and in presence of E6 and E6 and p53. At present, it is painstaking to reconstitute all these contacts by using the xcel files. It would be very informative to have such an overview on the main figures. This would nicely complement the plots such as the one in fig1 B,C and D, which are beautiful as an overview but do not inform on the precise lysine residues that are involved.

3) Production of E6 protein. I understood that E6 protein is produced as GST fusion and that it was not separated from GST in the experiments. I understood this, but it is not very clear in the paper. One has to read very carefully the mat meth to understand this. It would be nice if the text and the figures would mention that this is GST-E6 and not just E6.

4) GST is a dimer. Therefore the GST-E6 protein in this experiments is at least dimeric. Furthermore, HPV16 E6 is extremely cysteine-rich (14 cysteine residues for 151 residues in total), therefore highly prone to disulfide bridging, which also promotes self-association. Finally, the peptide-binding hydrophobic pocket of E6 also promotes aggregation. Indeed, in our own experience we never managed to obtain monodisperse samples of wild-type HPV16 E6. The X-ray structure of HPV16 E6 could be solved only at the expense of mutating four cysteine residues and providing E6 with its cognate target peptide (Zanier et al., Science 2013). One can therefore

expect that the wild-type GST-E6 protein used in the present manuscript was, at least in part, oligomeric. May the oligomeric state of E6, as well as its capacity to establish disulfide bridges, influence the outcome of the cross-linking experiments? This would deserve to be discussed.

Dr Gilles TRAVE, CNRS, Strasbourg, France.

Reviewer #2 (Remarks to the Author):

Sailer et al. aim at characterizing the structure and dynamics of the E6AP/UBE3A-E6-p53 complex. Although a partial structure of the complex has been determined before, it contained only a short peptide of E6AP. The partial structure did not allow determining how the catalytic HECT domain of E6AP is brought towards p53 substrate and how E6 stimulates the activity of E6AP. Since apparently high-resolution structure determination of the complex with the full-length E6AP is challenging, the authors used crosslinking mass spectrometry (XL-MS) and subsequent crosslink-driven structure modeling.

First, using both "standard" XL-MS and recently developed quantitative XL-MS, the authors observe that crosslink patterns within E6AP change upon E6 binding. Based on this, they propose that the N- and C-terminal domains of E6AP are in close proximity within the E6AP-E6 complex. Second, they address whether E6 promotes homo-oligomerization of E6AP as previously proposed. Since standard XL-MS overall cannot distinguish between inter- and intra-subunit crosslinks for homo-oligomers, they introduce a "SILAC-XL-MS" approach to distinguish these crosslinks types based on isotope labelled subunits. Based on this experiment, they find that E6 does not significantly promote homo-oligomerization. Finally, they utilize the collected crosslinks and available crystal structures of the partial complex and individual domains to model the structures of E6AP-E6 and E6AP-E6-p53 complexes using integrative modeling methodology. The obtained models are definitely not atomic resolution but authors address it by interpreting the resulting subunit localization probabilities rather than single atomic conformations.

Overall, the results were used to conclude that E6 brings together AZUL and HECT domains of E6AP and in the E6AP/UBE3A-E6-p53 complex both E6 and p53 interact with the HECT domain, which bears the catalytic center of E6AP. This then suggests that E6 stimulates E6AP activity by stabilizing a high-activity conformation of E6AP.

Overall, I have several positive comments but also major concerns.

The manuscript represents a very thorough XL-MS analysis and utilizes the best of XL-MS. It is remarkable what information can be now derived when state-of-the-art XL-MS methods are used together, and interpreted with modeling. The XL-MS methods are described appropriately, including thresholds used. XL-MS data is presented appropriately, relevant scores are included, crosslinks with associated scores are available in the supplement, and some known limitations of XL-MS taken into account. It includes an excellent application of XL-MS method to a system clearly requiring approaches such as XL-MS. However, I still have some major concerns, which I hope can be addressed.

Major concerns:

- From the Figure 2 one can clearly see that both monolinks and crosslinks increase abundance within the NTD. Likewise, both monolinks and crosslinks decrease within the central region of the protein. Although the fold change of the monolinks is low, the monolinks are changing consistently within these regions. This is a major concern, as it may indicate that, after all, the reactivity of the lysine residues does change upon addition of E6, and the changes in crosslinking patterns may be due to changed lysine reactivity. This could indicate, for example, opposite conclusions about the dynamics to the ones postulated in this paper and propose a model in which the AZUL domain is

close to, or interacts with, the HECT domain already in the apo-A6EP and upon interaction with E6 with the central region and the HECT domain, the central region gets buried by binding to E6 or to E6AP-CTD (decreasing accessibility of the lysines and decreasing both mono and intralinks), while the NTD becomes more exposed to the solvent (resulting in an increased accessibility of the lysines for both mono and intralinks). This would fit observations from the crystal structure that the E6-binding motif of E6AP is in the central region of E6AP. Also, many crosslinks between N- and C-terminus are observed even without E6 suggesting that AZUL and HECT domains are close to each other already without E6.

- How does the monolinks fold changes look for E6_L50E? Do they change similarly to E6_wt (increase in NTD, decrease in the central region)? If they look different, it would support that the monolink changes are due to interactions with E6. This information should be added to Supp. Figure 2 and Supp Tables.

- If the structure of AZUL domain is known, why has not it been included in the modeling? Could you run a test and model AZUL-HECT interaction independently based on crosslinks from E6AP and E6AP+E6 samples and compare if you get different results?

- The E6AP – E6 modeling is based on only 4 crosslinks, and E6AP – E6 – p53 modeling based on 7 crosslinks. It is not much in absence of complementary data. How would the results change if some of the crosslinks are excluded? How would the results change if you use Id-score threshold of 30?

- The confidence in the model and the hypothesis regarding the dynamics would significantly benefit from complementary data. Very likely SAXS experiments could provide valuable supporting information and actually could be also used in modeling, since the modeling framework used by authors includes SAXS restraint.

Minor comments

- It was very confusing to learn only in the Discussion that the crystal structure of the partial complex is available. The paragraph starting from “Recently, the crystal structure of the HPV-16 E6 oncoprotein bound to an E6AP- derived peptide of 12 amino acids containing the LQELL motif and to the DNA binding domain of p53 was solved”, which describes the fact and motivation for full length study, should be moved to the introduction.

- Isn't the discussion of disappearing crosslinks in first section redundant and not necessary because this is discussed again and much better based on q-XL-MS?

- Please explain in the Methods what a 'biological replicate' means in this study. Independent protein preparations?

- In “(...) leading to diminished crosslinking within the central region of the protein (Fig. 1C)” it is not clear which region and crosslinks exactly are exactly meant. Maybe indicate the “central region” on the figure and the changing crosslinks by another color? Color the disappearing crosslinks in both panel 1B and D?

- In “This technique is a recent innovation ref: 42” could a more specific reference be used instead of the cited review article? Perhaps: <https://www.ncbi.nlm.nih.gov/pubmed/23541715/> ?

- Include fold changes of monolinks in the Table S4. For example, the monolink fold changes within the C-terminus of E6AP are hard to read from the figure 2

- Page 9, top paragraph, Fig 3B and Fig 3C is referenced instead of Supp Figure 3.

- In Discussion, you mention that p53 is a bit too far for ubiquitylation but another copy of p53 could be close enough if the tetramer is bound. Isn't the p53 tetramer structure known? Could you test this hypothesis by superposing the tetramer structure onto p53 in your model?

- “we attained a first structural and functional understanding of the dynamics of the full-length E6AP-E6-p53 enzyme substrate complex” – is it really the “first”? Is it really “understanding”, given all the remaining questions mentioned by the authors (e.g. distance of p53 to the active site)? It is unnecessary overstated and could be changed to “we attained structural and functional insights into the dynamics of the full-length E6AP-E6-p53 enzyme-substrate complex”

- The complex is small and could be addressed by high-resolution modeling/docking such as Haddock or Rosetta instead of the coarse-grained modeling used. This would add a possibility of using physical potential and scoring to aid modeling. One could argue that the other methods do not utilize Bayesian crosslinking restraint, but the benefit of this restraint is not clear in this work. Could the authors try to compare their model to a model obtained using Haddock webserver,

which should be straightforward to run for this case?

- What is the benefit of using the Bayesian scoring function here? Were the weights of cross-links optimized as nuisance parameters? Was the empirical expected length distribution modeled with the priors?
- I believe the recently published method for assessing the sampling exhaustiveness and precision should be used, as it does not depend on an arbitrary clustering threshold (Viswanath et al., <https://www.ncbi.nlm.nih.gov/pubmed/29211988>)
- Is it possible to make adaptation of xQuest software suite for SILAC-XL-MS publicly available? This may be very useful to the community
- The model, including the localization densities, should be deposited in PDB-dev <https://pdb-dev.wwpdb.org/> or similar repository, or as supplement to the manuscript
- The raw MS data should be deposited in the PRIDE database <https://www.ebi.ac.uk/pride/archive/> rather than be available from authors on request
- Figure 5 and 6 – crosslinks are not clearly visible, could you make the lines thicker and indicate residue numbers
- Supplementary Fig. 3 –second b in the legend should be c
- Methods: “For our modeling runs, we employed the known crystal structures of the HECT domain of E6AP (PDB ID: 1c4z) and E6/p53 (PDB ID: 4xr8).” Do you mean that LOELL of E6AP has been excluded from 4xr8? Why? Wouldn't it provide useful constraint for modeling?
- Methods: was xQuest/xProphet FDR estimation used? What was the FDR cutoff for selecting crosslinks? Could you also include the FDR values of all the crosslinks in the tables? Even better, could you upload complete xQuest tables with all the columns?

Reviewer #3 (Remarks to the Author):

This paper nicely elucidated the structural dynamics and potential binding modes of the E6AP/UBE3A-E6-p53 using quantitative XL-MS technique in combination with integrative structural modeling. While I find the overall results are good, I have the following technical concerns and suggestions I would very much like the authors to address.

1. In part 1 of the result “Crosslinking of E6AP in the absence and presence of the E6 oncoprotein reveals distinctive crosslink patterns”, the authors show possible conformation change of E6AP upon binding to E6, as illustrated by diminished crosslinks within the central region of E6AP. By merely visualizing the crosslink pattern shown in Figure 1A, B and C, I could see two crosslinks are missing within the central region of E6AP when binding to E6. However, in part 2 of the results, “SILAC-XL-MS reveals weak interactions between N-terminal regions of E6AP, but not the HECT domain”, these two significantly changed crosslinks did not appear in Figure 2B. I wonder if they were not identified in this dataset or was not able to quantify due to missing values. I think the authors should give reasons for this issue explicitly.
2. In connection with my previous question, I think the authors should also give more explanations of their quantitative pipeline on if “match between runs” are performed and how missing values are handled (simply throw them away or imputation?).
3. Furthermore, if quantitative information is already obtained (part 2 of the result), I find it is pointless to present only identification with exactly the same sample using a whole section in the result (part 1). So I think the authors should either emphasize what are the additional information the readers can get from part 1 or they should completely remove part 1.
4. Also in Figure 2, I think it's more visually appealing to present an additional volcano plot to show the distribution of fold-changes of all peptides, mono-links as well as crosslinks in two different conditions (E6AP, and E6AP bound to E6) and highlight the cross-links selected as significant ones.
5. In part 3 “SILAC-XL-MS reveals weak interactions between N-terminal regions of E6AP, but not the HECT domain.”, the authors presented SILAC-XL-MS workflow to study the binding interface of homoligomers. It is indeed a very important question that urgently need to be addressed. However, I found it is difficult to believe treating the stable dimer with 6M Urea to denature it and

reforming the dimer by dialysis does not affect the protein structure and/or dimer formation. I guess the stability of the protein probably vary case by case, which I could not judge in the study of the authors. Nevertheless, I think the authors should provide additional information to further explain the following points: 1. Explain in a better way how the experiment was performed. I read that GST was treated with 6M urea but how E6AP was processed was not mentioned. 2) If E6AP was also treated with 6M urea, I think the author should at least provide some supporting information on how reliable the dimer re-formation is for E6AP, after treating with 6M Urea.

6. Also in part 3, the authors should provide information on how many crosslinks they find in their SILAC-XL-MS experiments in the main text and Figure 3, and make a proper comparison with the numbers in part1 and part2. I would like to see if SILAC combined XL-MS may affect the efficiency of crosslink identification.

7. "However, it is interesting to note that none of the detected interlinks is located within the C-terminal half of E6AP comprising the HECT domain, where the oligomerization site has been previously proposed to be located. In summary, our SILAC-XL-MS data suggests that E6AP can engage in homo-oligomeric interactions, but these are not mediated through the HECT domain, and that oligomerization is not initiated or intensified by E6." No crosslinker detected at the HECT domain does not necessarily mean no binding in this region. This sentence should be rephrased.

8. Lastly, I would like to give a general comment of the supplementary tables. Merely listing all cross-links in different experiments is tedious, and very difficult for the readers to seek for useful information. I suggest the authors change the way of presenting crosslinks to format that is similar to typical quantitative proteomics experiments, listing all identified crosslinks, their normalized intensities (indicating missing values), across all biological replicates and conditions in comparison, as well as t-test significance and p-value. In this way, it is straightforward for the readers to immediately see the quality of the data and what cutoff the authors are chosen to report the significance.

Point-by-point responses to reviewers' comments on "Structural dynamics of the E6AP/UBE3A-E6-p53 enzyme-substrate complex" (NCOMMS-18-11592)

Reviewer #1 (Remarks to the Author):

In HPV-induced cancers, the E6 HPV oncoprotein (~150 residues) primes the E6AP HECT ubiquitin ligase (~850 residues) to poly-ubiquitylate tumor suppressor p53 (~390 residues). It is known that the binding to E6 strongly activates the ubiquitin ligase activity of E6AP, not only towards p53 but also to other substrates of E6AP, including E6AP itself. A prior crystallographic structure has shown that full-length E6 protein interacts with a helical "LxxLL" motif derived from the central region of E6AP to recruit the "core" central domain of p53 (Martinez-Zapien et al., Nature 2016). However, high-resolution structural data concerning full-length E6AP alone or complexed to full-length E6 and/or full-length p53 are still missing.

In this manuscript, Sailer et al have applied state-of-the-art cross-linking mass spectrometry (XL-MS) to map 1) the intramolecular contacts between different regions of E6AP in the absence of E6 and p53; 2) the intramolecular contacts between different regions of E6AP in the presence of E6 and of E6 and p53; 3) the intermolecular contacts between different regions of two different molecules of E6AP in the absence of E6 and p53; 4) the contacts established by full-length E6AP with E6 and p53 within pre-formed E6-E6AP and E6-E6AP-p53 complexes. They have produced highly reproducible data which allow them to map the different contacts listed above, and extract exciting information about the conformational changes occurring to E6AP upon E6 binding, which may explain the resulting "hyper activation" of the E6AP enzymatic activity. Furthermore, based on the cross-linking patterns they compute distance-driven models of the relative spatial arrangements of the different domains of E6, E6AP, p53 and E2 ubiquitin ligase UbcH7 in a quaternary complex of these different proteins.

The experimental data are very convincing, and as far as I can judge (not being an expert in mass spectrometry approaches) extremely carefully performed and analyzed. The conclusions are very sound and convincing and will represent a very important contribution to our understanding of the fascinating mechanism of E6-E6AP mediated degradation of p53. My comments here will be only related to my field of expertise (biochemistry and structural biology).

We thank the reviewer for considering our experimental data as "very convincing", and our conclusions as "very sound and convincing and (...) a very important contribution to our understanding of the fascinating mechanism of E6-E6AP mediated degradation of p53".

1) It would be useful to represent (supplemental figure the sequences of the three proteins used in the study enlightening all lysine residues, with a different color for the lysines which have reacted and the ones which did not react.

We thank the reviewer for his suggestion and have added a Figure (**new Supplementary Figure 1**) where we highlight all lysine residues in E6AP, E6 and p53, for which crosslinks were identified. To account for this additional figure, we amended the text on page 6 as follows: "We applied isotopically labeled disuccinimidyl suberate (DSS) to crosslink highly purified E6AP (Fig. 1A) and

determined the crosslinking pattern within full-length E6AP (**for a general overview of the susceptibility of lysine residues to crosslinking in this study, see Supplementary Fig.1**).

2) *It would be very useful to include in the figures additional tables in the same format as the tables shown in Fig 4B and Fig 5B, to represent E6AP to E6AP contacts in the absence and in presence of E6 and E6 and p53. At present, it is painstaking to reconstitute all these contacts by using the xcel files. It would be very informative to have such an overview on the main figures. This would nicely complement the plots such as the one in fig1 B,C and D, which are beautiful as an overview but do not inform on the precise lysine residues that are involved.*

Thank you for pointing this out. We have added a table containing all E6AP to E6AP contacts in the absence and in the presence of E6 (wt), E6_L50E and p53. Due to space constraints, we have made an additional Supplementary Table (**new Supplementary Table S4**), where we show this information. For the sake of completeness, the overview in Supplementary Table S4 also contains E6 and p53 intra-protein crosslinks and the identified inter-protein crosslinks.

We have also added the following sentence to the manuscript on page 6: “In the presence of E6_L50E, we identified 131 unique lysine-lysine contact sites, which correspond to 325 unique crosslinks identified in total over 3 biological replicates (Supplementary Table 3; **for a direct comparison of all crosslinked peptides under the various conditions, see Supplementary Table 4**)”.

3) *Production of E6 protein. I understood that E6 protein is produced as GST fusion and that it was not separated from GST in the experiments. I understood this, but it is not very clear in the paper. One has to read very carefully the mat meth to understand this. It would be nice if the text and the figures would mention that this is GST-E6 and not just E6.*

We agree with the reviewer and have amended the manuscript accordingly, as follows:

Page 6: “We next repeated the experiment in the presence of the HPV-16 E6 oncoprotein (E6; note that a GST-E6 fusion protein was employed), identifying...”

Page 6: “To corroborate that changes in the crosslinking pattern were caused by interaction between E6AP and E6, we repeated the experiment using the L50E mutant of E6 (E6_L50E; a GST fusion protein was employed).

Figure 1: Panel Figure 1A was changed to “**GST E6 wt**” and “**GST E6_L50E**” and to the Figure legend (page 32): **(a)** SDS-polyacrylamide gel showing preparations of E6AP, wild-type HPV-16 E6 (E6 wt), and an E6 mutant that does not bind to E6AP (E6_L50E) used for XL-MS experiments (**E6 proteins were expressed as GST fusion proteins**).

Supplementary Fig. 7 (formerly Supplementary Fig. 5) figure was changed to “**GST E6 wt**” and “**GST E6_L50E**” and in the caption it was indicated that a GST-E6 fusion protein was used.

4) GST is a dimer. Therefore the GST-E6 protein in this experiments is at least dimeric. Furthermore, HPV16 E6 is extremely cysteine-rich (14 cysteine residues for 151 residues in total), therefore highly prone to disulfide bridging, which also promotes self-association. Finally, the peptide-binding hydrophobic pocket of E6 also promotes aggregation. Indeed, in our own experience we never managed to obtain monodisperse samples of wild-type HPV16 E6. The X-ray structure of HPV16 E6 could be solved only at the expense of mutating four cysteine residues and providing E6 with its cognate target peptide (Zanier et al., Science 2013). One can therefore expect that the wild-type GST-E6 protein used in the present manuscript was, at least in part, oligomeric. May the oligomeric state of E6, as well as its capacity to establish disulfide bridges, influence the outcome of the cross-linking experiments? This would deserve to be discussed.

Thank you for pointing out that under the conditions used, E6 is likely to be present - at least partially - in a multimeric form. To unambiguously address the possibility that the crosslinking data are affected by the multimerization status of E6, we would need to repeat the experiments using the E6 mutant mentioned by the reviewer, in which the cysteine residues were replaced by serine. However, since we do not know what the physiologically relevant situation is within a cell, the implications of results obtained in such an experiment would be unclear (if the results would indeed be different).

More importantly, we consider that an important physiological role for E6 dimerisation is unlikely for the following reasons:

First, in the experiment shown in Fig.3, we compared the potential oligomerization behaviour of E6AP in the absence and in the presence of E6 by determining the number of interlinks between different E6AP protomers and their position within E6AP. This revealed that E6 has no significant influence on the interlink pattern. This, in our opinion, clearly indicates that the oligomerization status of E6 does at least not affect the oligomerization behaviour of E6AP.

Secondly, all of our enzymatic assays (e.g. E6AP auto-ubiquitylation, E6-E6AP-mediated ubiquitylation of p53) are performed under similar conditions. If disulfide bond formation between E6 and E6AP would be a critical issue, it would likely result in deformation and inactivation of E6AP.

Thirdly, one of the main conclusions of our data is that in addition to the known interaction site, E6 comes into close proximity to the C-terminal region of the HECT domain. This could indeed be explained by either two E6AP binding sites on one E6 molecule or by binding of two E6 molecules to one E6AP molecule. Although we cannot distinguish between these possibilities, the crosslinks clearly show that two different regions of E6 are involved in the interaction with the known E6 binding site and with the HECT domain. Thus, it is at least possible that one E6 molecule interacts with one E6AP molecule.

Nevertheless, we have amended the manuscript to clarify these topics in the following points: On page 14 the following sentence was added: "While it cannot be excluded that the GST E6 fusion protein used in this study was, at least in part, oligomeric, this also clearly indicates that the oligomerization status of E6 does not affect the oligomerization behaviour of E6AP."

On page 15 the following sentences were added: "This could in principle be explained by two E6AP binding sites on one E6 molecule or by binding of two E6 molecules to one E6AP molecule. Although we cannot distinguish between these possibilities directly, the crosslinks identified

clearly show that two different regions of E6 are involved in the in the interaction with the known E6 binding site and with the HECT domain. It is therefore likely that one E6 molecule interacts with one E6AP molecule.”

Reviewer #2 (Remarks to the Author):

Sailer et al. aim at characterizing the structure and dynamics of the E6AP/UBE3A-E6-p53 complex. Although a partial structure of the complex has been determined before, it contained only a short peptide of E6AP. The partial structure did not allow determining how the catalytic HECT domain of E6AP is brought towards p53 substrate and how E6 stimulates the activity of E6AP. Since apparently high-resolution structure determination of the complex with the full-length E6AP is challenging, the authors used crosslinking mass spectrometry (XL-MS) and subsequent crosslink-driven structure modeling.

First, using both “standard” XL-MS and recently developed quantitative XL-MS, the authors observe that crosslink patterns within E6AP change upon E6 binding. Based on this, they propose that the N- and C-terminal domains of E6AP are in close proximity within the E6AP-E6 complex. Second, they address whether E6 promotes homo-oligomerization of E6AP as previously proposed. Since standard XL-MS overall cannot distinguish between inter- and intra-subunit crosslinks for homo-oligomers, they introduce a “SILAC-XL-MS” approach to distinguish these crosslinks types based on isotope labelled subunits. Based on this experiment, they find that E6 does not significantly promote homo-oligomerization. Finally, they utilize the collected crosslinks and available crystal structures of the partial complex and individual domains to model the structures of E6AP-E6 and E6AP-E6-p53 complexes using integrative modeling methodology. The obtained models are definitely not atomic resolution but authors address it by interpreting the resulting subunit localization probabilities rather than single atomic conformations.

Overall, the results were used to conclude that E6 brings together AZUL and HECT domains of E6AP and in the E6AP/UBE3A-E6-p53 complex both E6 and p53 interact with the HECT domain, which bears the catalytic center of E6AP. This then suggests that E6 stimulates E6AP activity by stabilizing a high-activity conformation of E6AP.

Overall, I have several positive comments but also major concerns.

The manuscript represents a very thorough XL-MS analysis and utilizes the best of XL-MS. It is remarkable what information can be now derived when state-of-the-art XL-MS methods are used together, and interpreted with modeling. The XL-MS methods are described appropriately, including thresholds used. XL-MS data is presented appropriately, relevant scores are included, crosslinks with associated scores are available in the supplement, and some known limitations of XL-MS taken into account. It includes an excellent application of XL-MS method to a system clearly requiring approaches such as XL-MS. However, I still have some major concerns, which I hope can be addressed.

We thank the reviewer for their comments and are happy to hear that she/he considers our manuscript an “excellent application of XL-MS method” which “represents a very thorough XL-MS analysis and utilizes the best of XL-MS”.

Major concerns:

- From the Figure 2 one can clearly see that both monolinks and crosslinks increase abundance within the NTD. Likewise, both monolinks and crosslinks decrease within the central region of the protein. Although the fold change of the monolinks is low, the monolinks are changing consistently within these regions. This is a major concern, as it may indicate that, after all, the reactivity of the lysine residues does change upon addition of E6, and the changes in crosslinking patterns may be due to changed lysine reactivity. This could indicate, for example, opposite conclusions about the dynamics to the ones postulated in this paper and propose a model in which the AZUL domain is close to, or interacts with, the HECT domain already in the apo-A6EP and upon interaction with E6 with the central region and the HECT domain, the central region gets buried by binding to E6 or to E6AP-CTD (decreasing accessibility of the lysines and decreasing both mono and intralinks), while the NTD becomes more exposed to the solvent (resulting in an increased accessibility of the lysines for both mono and intralinks). This would fit observations from the crystal structure that the E6-binding motif of E6AP is in the central region of E6AP. Also, many crosslinks between N- and C-terminus are observed even without E6 suggesting that AZUL and HECT domains are close to each other already without E6.

While we agree with the reviewer that it is very important to consider all possible explanations of the data, we do not share their concerns on these points.

In order to further test whether the changes in crosslinking patterns are indeed likely caused by conformational changes within E6AP due to binding of E6 or may be due to changed lysine reactivity (though it is not obvious to us how the addition of any protein could cause a general change in lysine reactivity), we have additionally calculated the fold changes for monolinks for the binding deficient E6_L50E mutant (**new Supplementary Figure 4A, Supplementary Table 6**). Quantification of the change in abundance of identified crosslinks within E6AP, when incubated with the binding-deficient E6_L50E mutant vs. E6AP alone, reveals a similar pattern for the fold change of monolinks as for wild-type E6 (Supplementary Figure 4A), but no significant up- or down-regulated intralinks (Supplementary Figure 4B), strongly indicating that the identified changes in the E6AP crosslinking pattern (Figure 2) are *not* due to a E6-mediated change in lysine reactivity independent of protein-protein interaction.

It is also worth noting that, due to lack of structural data, the boundaries of a 'NTD' within E6AP cannot be clearly defined. The only structural information available before the HECT domain is for the AZUL domain (spanning the first 60 AAs of E6AP). Therefore, it is generally assumed that the N-terminal region of E6AP comprises everything until the HECT domain (i.e. the first 494 AAs). Applying these boundaries for the 'NTD' we see approximately equal up- and down-regulation of monolinks within this region. Moreover, looking at the monolink pattern within the binding-deficient E6_L50E mutant, we see both up- and down-regulated links, even within the very N-terminal part of E6AP (e.g. the first 200 amino acids) (Supplementary Figure 4A).

Furthermore, the extent of up- and down-regulation of monolinks is, at least for some links, orders of magnitudes below the ones observed for the intralinks, arguing against solvent exposure being the common reason underlying this change. The change in monolink pattern falls within a range which represents in our view the 'natural fluctuation' (i.e. noise) of a quantitative XL-MS

experiment' (as commonly used in 'conventional proteomics' experiments) and in our view this indicates that one should generally refrain from extracting trends from data that lie within these margins (this was the reason why we have introduced these margins of significance for a quantitative XL-MS experiment in the first place).

Regarding the issue of AZUL and HECT domain proximity prior to addition of E6, we do not claim that the AZUL and HECT domains are not close to each other already in the absence of E6, but rather that they must come into **closer** proximity upon binding of E6 and/or that binding of E6 stabilizes such a conformational state. The reviewer rightly points out that several crosslinks between the AZUL and HECT domains are already identified in the apo state. Given the crosslinker used (DSS H₁₂/D₁₂) and the chemistry of the crosslinking reaction, this suggests that linked residues in the AZUL and HECT domain are separated by no more than approximately 30 Å (Erzberger et al., Cell, 2014, PMID: 25171412) (unless the crosslinked sites are situated in very flexible regions). We also agree with the reviewer that the detected decrease in the "central region" in both mono and intralinks could be explained by "decreasing accessibility of the lysines (by) binding to E6". This is exactly in line with our experimental data, where we observed crosslinks between E6 and the known binding site located in the "central region" of E6AP but also with the HECT domain and which we interpreted – supported by our integrated modelling data – as binding of E6 to E6AP via these two sites. A tight binding is expected to generally obstruct accessibility to subjacent lysine residues – not discriminating between monolinks or Intralinks. Our line of argument is that a diverging pattern in monolink and intralink abundance provides further evidence of a **conformational change** in this region – as one can see in our data for the very N-terminal part (the N-terminal 350 amino acids roughly) and the C-terminal end of the HECT domain.

In summary, there is no indication that the addition of E6 has any influence on the general reactivity of lysine residues (except within regions that E6 directly binds) and - in particular when data for the binding deficient E6_L50E mutant is taken into account - there is no significant trend in terms of monolink abundances in the N-terminal part of E6AP (under any definition). The data therefore clearly indicates that the identified changes in the E6AP crosslinking pattern are not due to changed lysine reactivity, but most likely due to a conformational change that brings the N- and C-termini of E6AP into closer proximity. Nevertheless, we thank the reviewer for prompting us to more rigorously explain our reasoning on this point, and we have modified the text in Supplementary Fig. 4 (formerly Supplementary Fig. 2) – see next point.

- How does the monolinks fold changes look for E6_L50E? Do they change similarly to E6_wt (increase in NTD, decrease in the central region)? If they look different, it would support that the monolink changes are due to interactions with E6. This information should be added to Supp. Figure 2 and Supp Tables.

As abovementioned, these changes are now shown graphically in the new Supplementary Figure 4A. The significantly changed monolinks in the central region are absent in the mutant E6_L50E dataset, consistent with the notion that decreased monolinking within this region is caused by E6 binding. As for the wild-type E6 dataset, no monolinks within the N-terminal region change

significantly in the L50E dataset, although in this case monolinks are both up- and down-regulated within the N-terminal region (Supplementary Figure 4A).

As suggested by the reviewer, we have added the fold changes for the monolinks using the binding-deficient E6_L50E mutant vs. E6AP alone in Supplementary Figure 4A and Supplementary Table 6. While the figure caption has been changed to: Quantification of the change in abundance of identified crosslinks within E6AP, when incubated with the binding-deficient E6_L50E mutant vs. E6AP alone, shows no significantly up- or down regulated monolinks and reveals overall a similar fold change pattern for monolinks as for wild-type E6, even though the significantly changed monolinks in the central region are absent in the mutant E6_L50E dataset, consistent with the notion that decreased monolinking within this region is caused by E6 binding (**A**). As monolinks are both up and down-regulated throughout the N-terminal region in the E6_L50E mutant, it is difficult to conclude with confidence what is causing the cluster of slightly positively regulated monolinks between amino acids ~70-130 in the wild-type dataset, but the basically complete absence of significantly up- or down-regulated intralinks (**B**), strongly indicates that the identified changes in the E6AP intralink pattern, as observed in the experiment to Figure 2, are caused by binding of E6

- If the structure of AZUL domain is known, why has not it been included in the modeling? Could you run a test and model AZUL-HECT interaction independently based on crosslinks from E6AP and E6AP+E6 samples and compare if you get different results?

Although a high-resolution structure for the AZUL domain is available, its position relative to the HECT domain is unknown. Therefore, we would need to ensure connectivity between the two domains via coarse-grained beads. In the case of the two domains, this amounts to ~450 amino acids represented as beads. **Figure 1** below shows one of the resulting density maps from an E6AP plus E6 modeling run, where all unknown regions were represented as beads including the AZUL domain. However, since there are regions without any crosslinked lysines in that sequence, we were not confident enough in the resulting density maps to include them in the manuscript.

Figure 1 Structural model of the binding interface between E6AP and E6.

- The E6AP – E6 modeling is based on only 4 crosslinks, and E6AP – E6 – p53 modeling based on 7 crosslinks. It is not much in absence of complementary data. How would the results change if some of the crosslinks are excluded? How would the results change if you use Id-score threshold of 30?

The E6AP-E6 modeling was actually based on 13 interlink crosslinks for the E6AP-E6 interface, and 19 interlinks for the E6AP-E6-p53 interface since unique crosslinked peptides (e.g. multiple unique peptides can link one unique crosslinking site) were used as input as modeling restraints.

As suggested by the reviewer, we have tested how using a crosslinking dataset, where 15 percent of the links were randomly removed (jackknifing), as well as limiting the dataset to links with an Id-score >30, would influence our modeling results.

Figure 2 shows the results of the original modeling run of the binding interface between the HECT domain of E6AP and E6 (A), the jackknife run where 15 percent of the crosslinks were randomly removed (B), and the run using only very high-scoring crosslinks (Id > 30) (C). Each figure contains the original structural alignment of the proteins overlaid with the density maps from the respective modeling runs. In both cases (jackknife and Id>30), the sampling runs converged onto the same clusters with precisions within 1 Å of the original runs.

Figure 2: Structural model of the binding interface between the HECT domain of E6AP and E6. Note that while going through our log files, we noticed a small error on our part. For assigning the amino acids from the pdb file to the fasta sequence of E6, we used an incorrect offset of one amino acid. This led to all amino acids of E6 in the internal model of IMP being shifted by one. We therefore redid all our modeling runs without this offset. Importantly, no perceivable change in the density map was observed, except for a small translational shift. Not surprisingly, the center model in the modeling run not containing p53, which we used to align the pdb structure to, shifted also only marginally (as even in the same modeling runs, the center model can shift a few angstroms between repeated runs). Importantly, also here the general orientation remained the same. The center models as well as the density maps of the runs containing p53 showed no discernible shift by the offset. Also the distance between the catalytic cysteine at position 820 of E6AP and lysine 292 of p53 remains the same.

The same procedure was repeated for the structural model of the binding interface between the HECT domain of E6AP, E6, and p53. Also in this case did the sampling runs converge onto the same clusters with precisions within 1 Å of the original runs (**Figure 3**).

Figure 3: Structural model of the binding interface between the HECT domain of E6AP, E6, and p53. Shown are the results of the original modeling run of the binding interface between the HECT domain of E6AP, E6, and p53 (A), the jackknife run where 15 percent of the crosslinks were randomly removed (B) and the run using only very high-scoring crosslinks ($I_d > 30$) (C). Each figure contains the original structural alignment of the proteins overlaid with the density maps from the respective modeling runs.

We therefore added to the method section on page 24 of the manuscript the following paragraph: “To assess robustness, the modeling procedure described above was repeated while randomly removing 15% of the crosslinks (jackknifing). Additionally, we also repeated the modeling using only crosslinks with I_d -scores greater than 30. In both cases the sampling runs converged onto the same clusters with precisions within 1 Å of the original runs, as can also be seen in the rmsd matrices produced by these runs (Supplementary Fig. 6).”

Updated rmsd matrices have been included in Supplementary Fig. 6 (formerly Supplementary Fig. 4). We thank the reviewer for suggesting these validation approaches.

- The confidence in the model and the hypothesis regarding the dynamics would significantly benefit from complementary data. Very likely SAXS experiments could provide valuable supporting information and actually could be also used in modeling, since the modeling framework used by authors includes SAXS restraint.

Besides confirming the known interaction site between E6AP and E6, we identified an additional interface between E6AP and E6. We also show that this results in the positioning of E6 and p53 in the immediate vicinity of the catalytic center of E6AP. Our SILAC-XL-MS data also suggest that E6AP can engage in homo-oligomeric interactions, but that these are not mediated via the HECT domain, and that oligomerization is not initiated or intensified by E6. These main findings of our manuscript do not necessitate dynamic information and are already supported by multiple lines of evidence. Here, our modeling framework is merely used as an additional tool to validate our findings, which in principle we could also omit without losing critical information. The confidence

in these models was already very high (see method section integrative structural modeling), as we were able to revert to high-resolution structural data of the respective protein domains in this regard (i.e. crystal structures of the HECT domain and E6). Moreover, we were able to additionally increase the confidence in our models by performing additional modeling runs using only a subset of the data (jackknifing) or only a subset of high-confidence crosslinks (see above). We are therefore convinced that this core part of our findings, and thus also the models resulting from these, would not benefit from additional low-resolution data, as for example obtained by SAXS.

In regards to dynamics, we show that binding of E6 induces conformational rearrangements within E6AP that bring its N- and C-terminal regions into close proximity. We hope that our line of arguments addressing this point above and our additional data (Supplementary Figure 4A and Supplementary Table 6) already convinced the reviewer that this is likely the case. More importantly, SAXS data would need to be obtained from particles using full-length E6AP, E6 and potentially also p53. As we possess high-resolution structures only on a negligible part of the NTD of E6AP (i.e. 60 out of the nearly 500 N-terminal amino acids), it seems very unlikely that SAXS, which can provide data with a resolution in the range of 10 to 250 angstrom, would be able to significantly enhance our molecular understanding of the NTD of E6AP. This is also the reason, why we have deliberately refrained from using full-length E6AP for our integrated structural modeling approach (see also statement above regarding integrated modeling of the AZUL domain). Obtaining additional high-resolution data of the NTD of E6AP or full-length E6AP will likely increase our understanding, but full-length E6AP has so far proven to be not amenable to conventional structural analysis by crystallography or single-particle electron microscopy.

Finally, it is important to recapitulate that data on the dynamics of the N and C termini of E6AP bear no influence on our main finding, that is the positioning of E6 and p53 in the immediate vicinity of the catalytic center of E6AP and is also not required for our model of how E6 can both stimulate the ubiquitin ligase activity of E6AP and facilitate the transfer of ubiquitin from E6AP onto p53. While we therefore agree that the confidence in experimental results generally benefits from additional and complementary data, we believe that in this particular case, the additional information that could potentially be gained from SAXS experiments is likely to be negligible, and we therefore decided to not carry out SAXS experiments and hope that the reviewer agrees with our judgement.

Minor comments

- It was very confusing to learn only in the Discussion that the crystal structure of the partial complex is available. The paragraph starting from “Recently, the crystal structure of the HPV-16 E6 oncoprotein bound to an E6AP- derived peptide of 12 amino acids containing the LQELL motif and to the DNA binding domain of p53 was solved”, which describes the fact and motivation for full length study, should be moved to the introduction.

We apologize for this oversight and have moved the following sentence on page 4 of the introduction: “Similarly, the crystal structure of E6 bound to an E6AP-derived peptide of 12 amino acids containing the LQELL motif and to the DNA binding domain of p53 was recently solved ⁴¹. While revealing the role of the E6-binding region of E6AP in assembling the E6-p53 interaction,

the structure did not provide any insight into how full-length E6AP is oriented towards E6 and p53 in the E6AP-E6-p53 complex and how the E3 activity of E6AP becomes stimulated by interaction with E6.”

- Isn't the discussion of disappearing crosslinks in first section redundant and not necessary because this is discussed again and much better based on q-XL-MS?

While we agree that there is a certain overlap between the first and the second section of the manuscript, we feel that both sections are needed as it allows us to first focus on the global – non-quantitative – view what also includes the E6 oncoprotein in the picture before moving on to a more focussed analysis of the quantitative changes within E6AP. We are afraid that by merging/cutting these sections, we would lose critical information.

- Please explain in the Methods what a 'biological replicate' means in this study. Independent protein preparations?

Biological replicate in this study means independent protein preparations – i.e. separate batches of protein expression and purification were carried out on different days.

We have therefore added a clarifying statement in the method section on page 19: “Crosslinks were only considered during structural analysis, if they were identified in at least 2 of 3 biological replicates (i.e. **separately expressed and purified batches of protein**) with $\Delta S < 0.95$ and at least one Id score ≥ 25 . A list of all identified links can be found in Supplementary Excel File 1 (containing Supplementary Tables 1 – 10).”

- In "(...) leading to diminished crosslinking within the central region of the protein (Fig. 1C)" it is not clear which region and crosslinks exactly are exactly meant. Maybe indicate the "central region" on the figure and the changing crosslinks by another color? Color the disappearing crosslinks in both panel 1B and D?

As no defined structural region except for the already known ones (AZUL, HECT domain, HECT and E6 binding site) can be identified for E6AP without additional high-resolution data, we feel that it would potentially be misleading to define an exact “central region”. However, in order to be as precise as possible we have changed the text on page 6 to: “... leading to diminished crosslinking within the central region of the protein (i.e. **the region of E6AP between the AZUL and the HECT domain**) (Fig. 1C).”

- In "This technique is a recent innovation ref:42" could a more specific reference be used instead of the cited review article? Perhaps: <https://www.ncbi.nlm.nih.gov/pubmed/23541715/> ?

While the cited article provides a good overview over a specific workflow for quantitative XL-MS, it is not directly applicable to the workflow used in this study. In the cited manuscript, isotope-coded crosslinkers are used for quantification, while we use isotope-coded linkers mainly for identification. The respective specific reference would be Walzthoeni et al., Nat Methods. 2015

(PMID: 26501516). We had so far *not* cited this paper but rather a general review (which references to all quantitative XL-MS workflows and studies known to date) in order to illustrate and clarify that we are not the only group that contributes to the advancement of quantitative XL-MS, but happily added it to the revised manuscript.

- *Include fold changes of monolinks in the Table S4. For example, the monolink fold changes within the C-terminus of E6AP are hard to read from the figure 2*

The fold changes were already part of Supplementary Table S5 (formerly Supplementary Table S4) under the column "log2 ratio". For clarification, we have changed the heading of this column to "Fold Change (log2 ratio)". Supplementary Table S6 (formerly Supplementary Table S5) has been changed accordingly.

- *Page 9, top paragraph, Fig 3B and Fig 3C is referenced instead of Supp Figure 3.*

Thank you for pointing this out. We corrected the text accordingly.

- *In Discussion, you mention that p53 is a bit too far for ubiquitylation but another copy of p53 could be close enough if the tetramer is bound. Isn't the p53 tetramer structure known? Could you test this hypothesis by superposing the tetramer structure onto p53 in your model?*

To our knowledge, the structure of a p53 tetramer has not yet been reported, and we are therefore unable to test this hypothesis by superposing its structure onto p53 in our model. (the Halazonetis group reported on the structure of a "p53 tetramer" bound to DNA; PMID: 21522129. However, the p53 version used consists of the oligomerization domain of p53 - amino acids 322-358 - directly fused to the DNA binding domain - amino acids 94-291. Thus, it contains two major deletions in the C terminus - amino acids 292-321 and amino acids 359-392. In addition, the DNA binding domains contains 13 (!) point mutations. Thus, the structure of this "p53 tetramer" has to be considered with caution and is likely not of use for our studies).

- *"we attained a first structural and functional understanding of the dynamics of the full-length E6AP-E6-p53 enzyme substrate complex" – is it really the "first"? Is it really "understanding", given all the remaining questions mentioned by the authors (e.g. distance of p53 to the active site)? It is unnecessary overstated and could be changed to "we attained structural and functional insights into the dynamics of the full-length E6AP-E6-p53 enzyme-substrate complex"*

We agree with the reviewer and have corrected the text accordingly.

- The complex is small and could be addressed by high-resolution modeling/docking such as Haddock or Rosetta instead of the coarse-grained modeling used. This would add a possibility of using physical potential and scoring to aid modeling. One could argue that the other methods do not utilize Bayesian crosslinking restraint, but the benefit of this restraint is not clear in this work. Could the authors try to compare their model to a model obtained using Haddock webserver, which should be straightforward to run for this case?

The reviewer is correct in pointing out that using physical potential and scoring may potentially help the modeling. However, since we only included regions with a known crystal structure in the final modeling runs, we decided that using the crosslinks as (bayesian) distance restraints in conjunction with the excluded volume restraint should suffice in our case. However, if the reviewer feels that additional high-resolution modeling/docking by Haddock/Rosetta is still needed to support our modeling results, we will carry those out.

- What is the benefit of using the Bayesian scoring function here? Were the weights of cross-links optimized as nuisance parameters? Was the empirical expected length distribution modeled with the priors?

The weights of the crosslinks were determined by using an isotonic regression, weighing them according to their Id-score as given by *xQuest*. This approach is beneficial as it allows us to include lower-scoring links (i.e. Id-score < 25) in the modeling runs, adjusting their weight according to their score, without classifying them manually according to their uncertainties. The empirical length of the crosslinker is taken into account by the forward model as described in the Extended Experimental Procedures in [Erzberger, 2014]. We defined the maximal length as 23 Å (two lysine side chains plus the spacer arm of the linker).

- I believe the recently published method for assessing the sampling exhaustiveness and precision should be used, as it does not depend on an arbitrary clustering threshold (Viswanath et al., <https://www.ncbi.nlm.nih.gov/pubmed/29211988>)

We thank the reviewer for mentioning the manuscript by Viswanath et al. The authors show indeed a great pipeline for optimizing and streamlining the clustering process. Especially finding the sampling precision in an automated fashion is an impressive feat. However, in the scope of our modeling we consider our clustering analysis sufficient. Each modeling run ran for 15,000 frames on 16 cpu cores. The lowest scoring models started to appear as early as within the first 15% of the total number of frames, which makes us reasonably sure that we explored modeling space sufficiently. Since we ran each modeling run in triplicate (and pooled the results for the final clustering), we were also able to cluster each replicate individually and to confirm that we receive the same clusters as we do in the pooled clustering. Also, our clustering threshold of 5 Å was not chosen arbitrarily, but rather after comparing it to thresholds of 10 Å and 15 Å and finding it the most appropriate threshold for having well defined clusters. While we cannot report a sampling precision in the fashion shown in the mentioned manuscript, we calculated the cluster precision using the RMSF as well. When looking at the cross-correlation (using UCSF Chimera) between

the densities of the clusters of the replicates compared to the pooled clusters, we find very high cross-correlation coefficients: [0.9962, 0.9978, 0.9923] for the E6 main cluster in the HECT and E6 run and [0.992, 0.9765, 0.984] for the E6 main cluster and [0.9812, 0.9955, 0.9694] for the p53 main cluster in the HECT, E6 and p53 run. While we do find the method described by Viswanath et al. highly relevant, we are confident that in our case, our analysis of the clustering is sufficient and appropriate.

- Is it possible to make adaptation of xQuest software suite for SILAC-XL-MS publicly available? This may be very useful to the community

— We appreciate that the reviewer considers our adaption potentially useful for the community and are happy to make our adaption publicly available. We have compiled a repository of all the required changes to the xQuest software. The git repository containing all modified xQuest files and instructions on how to install them may be found on github.com (https://github.com/stengellab/silac_xl_ms). We have added this information to the method section.

— *- The model, including the localization densities, should be deposited in PDB-dev <https://pdb-dev.www.pdb.org/> or similar repository, or as supplement to the manuscript*

We have deposited the structural model of the binding interface between the HECT domain of E6AP and E6 and the structural model of the binding interface between the HECT domain of E6AP, E6, and p53, in both cases including localization densities, in PDB-dev. The models are currently checked and we are waiting for a publicly accessible link.

- The raw MS data should be deposited in the PRIDE database <https://www.ebi.ac.uk/pride/archive/> rather than be available from authors on request

We have deposited the raw MS data in PRIDE (px-submission #277951; Submission Reference: 1-20180530-44244)

- Figure 5 and 6 – crosslinks are not clearly visible, could you make the lines thicker and indicate residue numbers

We have adjusted the thickness for the crosslinks in Figures 4C and 5C.

We have also tested to add the respective residue numbers as well but found that it not only decreased the overall visibility but also obstructed clarity of the respective figures. Yet, for exactly this reason, we had already added the tables containing the respective residue numbers in Figures 4B and 5B.

- Supplementary Fig. 3 –second b in the legend should be c

We have corrected the text accordingly.

- *Methods: "For our modeling runs, we employed the known crystal structures of the HECT domain of E6AP (PDB ID: 1c4z) and E6/p53 (PDB ID: 4xr8)." Do you mean that LQELL of E6AP has been excluded from 4xr8? Why? Wouldn't it provide useful constraint for modeling?*

As the LQELL peptide contains no lysines and could therefore not be crosslinked, it would have added no additional information to the modeling runs (except for volume exclusion). We therefore decided against including it into our modeling runs.

— *- Methods: was xQuest/xProphet FDR estimation used? What was the FDR cutoff for selecting crosslinks? Could you also include the FDR values of all the crosslinks in the tables? Even better, could you upload complete xQuest tables with all the columns?*

— *XProphet* was indeed used to calculate FDR estimations. However, as we have shown previously, Id-Scores provide a better measure to ensure comparability over multiple and independent datasets (Erzberger et al., Cell, 2014, PMID: 25171412). Id-Scores were therefore chosen as cut-off criterion. We also deliberately decided against uploading the complete xQuest tables, as in our experience many readers, in particular the ones not familiar with mass spectrometry, are side-tracked from the relevant information by the many additional columns, particularly the MS specific ones. However, complete xQuest and xProphet have been uploaded to PRIDE (px-submission #277951; Submission Reference: 1-20180530-44244).

We also want to thank the reviewer again for his or her very thorough and insightful comments.

Reviewer #3 (Remarks to the Author):

This paper nicely elucidated the structural dynamics and potential binding modes of the E6AP/UBE3A-E6-p53 using quantitative XL-MS technique in combination with integrative structural modeling. While I find the overall results are good, I have the following technical concerns and suggestions I would very much like the authors to address.

We thank the reviewer for appreciating our findings and results.

1. In part 1 of the result "Crosslinking of E6AP in the absence and presence of the E6 oncoprotein reveals distinctive crosslink patterns", the authors show possible conformation change of E6AP upon binding to E6, as illustrated by diminished crosslinks within the central region of E6AP. By merely visualizing the crosslink pattern shown in Figure 1A, B and C, I could see two crosslinks are missing within the central region of E6AP when binding to E6. However, in part 2 of the results, "SILAC-XL-MS reveals weak interactions between N-terminal regions of E6AP, but not the HECT domain", these two significantly changed crosslinks did not appear in Figure 2B. I wonder if they were not identified in this dataset or was not able to quantify due to missing values. I think the authors should give reasons for this issue explicitly.

We presume that the reviewer refers to the following links: 327:x:529; 300:x:779 (same holds true for 529:x:708; 327:x:412).

As the reviewer correctly points out, those links are only shown in Figure 1B but not in the quantitative dataset in Figure 2B (we assume that the reviewer is referring to “binding of E6 induces a conformational rearrangement of E6AP bringing N- and C-termini into closer proximity” and the list of significantly up or down regulated crosslinks in Figure 2 rather than to the list of links between E6AP protomers in “SILAC-XL-MS reveals weak interactions between N-terminal regions of E6AP, but not the HECT domain”). The reason is that those links were *identified* but not *quantified* according to our cut-off criteria. While only links that were identified with high-confidence (e.g. Id-Score of >25, deltaS Score <0.95 and identification in at least two out of 3 independent biological replicates) were *admitted* to our quantitation pipeline, only a subset of those links also passed our cut-off criteria for quantification (violations = 0; log2ratio $\geq \pm 1.5$ and a p-value of ≤ 0.01). At this point, it is also important to emphasize that while identification of crosslinks is happening on the level of the unique peptide (i.e. unique peptide sequence and linkage site), quantification is done on the level of the unique crosslinking site (uxID), i.e. multiple different crosslinked peptides can contribute to a unique crosslinking site. Therefore, a crosslink that has a violation of >0 or a p-value of $\geq \pm 0.01$ will not show up as a quantified link in Figure 2 (i.e. not even as a grey line). Only if the link additionally has undergone a fold change of log2ratio $\geq \pm 1.5$ between conditions will the link be considered significantly up- or down-regulated and is shown in green or red, respectively. In the particular case of the links mentioned above, the reason for their non-appearance in Figure 2 has been that they had violations >0; i.e.: 327:x:529 (2 violations); 300:x:779 (2 violations); 529:x:708 (1 violation); 327:x:412 (2 violations).

We have tried to discuss these points as explicitly as possible in the extensive method section of the manuscript.

2. In connection with my previous question, I think the authors should also give more explanations of their quantitative pipeline on if “match between runs” are performed and how missing values are handled (simply throw them away or imputation?).

A detailed description of our pipeline can be found in the accompanying paper to *xTract* (citation 45 of the manuscript: Walzthoeni et al., Nat Methods. 2015 (PMID: 26501516)) with the settings that we used as described in the method section of the manuscript. *XTract* indeed imputes missing values (similarly to the option “match between runs”) and assigns a fractional value for a crosslinking site that was identified only in one of the two (or multiple) samples in order to allow comparison also for crosslinks that are present only in one state. To clarify this point, we have added a column with “imputed values” to the respective datasets in Supplementary Tables S5 and S6.

In addition, we have uploaded complete *xQuest* and *xProphet* result files to PRIDE. This will allow to directly search for all crosslinks with higher log2ratios and p-values, and a quick re-run of *xProphet* using a higher “violations” setting will also quantify additional links carrying those higher values (we have not done this, as our experience shows that unique crosslinking sites with higher violations are generally not trustworthy in their assignment).

3. Furthermore, if quantitative information is already obtained (part 2 of the result), I find it is pointless to present only identification with exactly the same sample using a whole section in the result (part 1). So I think the authors should either emphasize what are the additional information the readers can get from part 1 or they should completely remove part 1.

As already stated above (response to reviewer 2), while we agree that there is a certain overlap between the first and the second section of the manuscript, we feel that both sections are needed as it allows us to first focus on the global – non-quantitative – view that also includes the E6 oncoprotein into the picture, before moving on to a more focussed analysis of the quantitative changes within E6AP. We therefore fear that by merging/cutting these sections we would lose critical information.

4. Also in Figure 2, I think it's more visually appealing to present an additional volcano plot to show the distribution of fold-changes of all peptides, mono-links as well as crosslinks in two different conditions (E6AP, and E6AP bound to E6) and highlight the cross-links selected as significant ones.

As suggested by the reviewer, we now also present the data as a volcano plot. While this widely accepted representation has many advantages, we still feel that in this particular case – i.e. the quantification of crosslinks in contrast to the more common quantification of peptides - the way we have chosen to present the data offers some unique benefits. In particular, it directly allows the visual connection between the upregulated crosslinking sites and their position within the protein. We therefore would like to stick to our current way of presenting the data in the main part of the manuscript. The Volcano plot is presented in the Supplemental Information (**new Supplementary Fig. 3**).

5. In part 3 “SILAC-XL-MS reveals weak interactions between N-terminal regions of E6AP, but not the HECT domain.”, the authors presented SILAC-XL-MS workflow to study the binding interface of homooligomers. It is indeed a very important question that urgently need to be addressed. However, I found it is difficult to believe treating the stable dimer with 6M Urea to denature it and reforming the dimer by dialysis does not affect the protein structure and/or dimer formation. I guess the stability of the protein probably vary case by case, which I could not judge in the study of the authors. Nevertheless, I think the authors should provide additional information to further explain the following points: 1. Explain in a better way how the experiment was performed. I read that GST was treated with 6M urea but how E6AP was processed was not mentioned. 2) If E6AP was also treated with 6M urea, I think the author should at least provide some supporting information on how reliable the dimer re-formation is for E6AP, after treating with 6M Urea.

We updated the methods to clarify that GST – but not E6AP – was treated with 6 M urea to induce dissociation of light-light and heavy-heavy dimers. As described in the method section, dissociated light and heavy GST monomers were mixed before urea was removed, to enable formation of light-heavy dimers prior to crosslinking, as GST dimerizes with a sub-nanomolar dissociation constant (Fabrini *et al.*, Biochemistry, 2009, PMID 19795889). In comparison,

oligomeric E6AP interactions are transient (see Supplementary Figure 4D, formerly 3D), so there was no need to dissociate light-light and heavy-heavy E6AP dimers prior to cross-linking by treatment with urea.

To clarify these issues, we amended the manuscript on page 19 and added the following sentences: “Purification of isotope-labeled E6AP was performed as described above for unlabeled E6AP. **As oligomeric E6AP interactions are transient** under the conditions used for purification (**Supplementary Figure 5D, formerly Figure 4D**), **there was no need to dissociate light-light and heavy-heavy E6AP dimers prior to cross-linking**”

and page 22: “Upon expression and purification, the two GST/E6AP preparations were mixed in a 1:1 ratio (prior to mixing, GST preparations were treated with 6 M urea to induce dissociation of light-light and heavy-heavy dimers, see above), and.....”.

Regarding the possibility that unfolding-refolding induced by treatment with urea may alter the conformation of dimeric GST: Hornby and co-workers (*Biochemistry*, 2000. PMID 11015213) performed an in-depth investigation of GST unfolding with either guanidine hydrochloride or urea. By monitoring second and tertiary structure as well as enzyme activity, they report that unfolding induced by 6 M urea is 100 % reversible. To confirm that treatment with urea did not affect the conformation of dimeric GST, we performed XL-MS with untreated light GST (**new Supplementary Table 12**). The crosslinking patterns for untreated GST and GST subjected to urea unfolding/refolding are highly similar. For example, the 19 highest scoring unique cross-links detected in untreated GST were also detected in urea-treated GST. Furthermore, the dimeric crosslink spanning lysines 10 and 124 is present in the top 3 highest scoring crosslinked peptides irrespective of urea treatment.

To clarify these issues, we incorporated this additional dataset for XL-MS with untreated light GST (**new Supplementary Table 12**) in our revised manuscript. We also amended the manuscript on page 19 and added the following sentences:

“Dissociated labeled and isotope-labeled monomers were then recombined, and urea was removed by serial dilution to enable formation of mixed unlabeled-isotope-labeled dimers. **To confirm that treatment with urea did not affect the conformation of dimeric GST, we performed XL-MS also with untreated light GST. The crosslinking patterns for untreated GST (Supplementary Table 12), and GST subjected to urea unfolding/refolding (Supplementary Table 7), are highly similar.**”

6. Also in part 3, the authors should provide information on how many crosslinks they find in their SILAC-XL-MS experiments in the main text and Figure 3, and make a proper comparison with the numbers in part1 and part2. I would like to see if SILAC combined XL-MS may affect the efficiency of crosslink identification.

We thank the reviewer for pointing out this oversight.

We have collected the data from the Supplementary Information (Supplementary Tables S1, S2, S9 and S10) and the respective numbers are:

E6AP: (no labelling): 145 uxIDs within full-length E6AP, corresponding to 396 unique crosslinks (SILAC): 109 uxIDs and 379 unique crosslinks (containing 204 light-light, 168 heavy-heavy and 7 light-heavy links)

E6AP plus E6: (no labelling): 146 uxIDs within full-length E6AP, corresponding to 408 unique crosslinks

(SILAC): 108 uxIDs and 383 unique crosslinks (containing 204 light-light, 172 heavy-heavy and 7 light-heavy links).

— The number of uxIDs is therefore lower in experiments, where SILAC was used in combination with crosslinking. This may possibly be explained by a direct effect of SILAC on the efficiency of crosslink identification. However, as the number of overall identified unique crosslinks is roughly the same in SILAC and non-SILAC crosslinking experiments, this effect may also be explained by the relative smaller amount of peptides that was loaded on the column (i.e. in both cases 1 ug of total peptide amount was loaded onto the LC MS/MS column; which should translate to half the amount of light E6AP peptides being present in the SILAC vs the non-SILAC sample). Additional experiments, which are however beyond the scope of this manuscript, would be needed to fully clarify this issue.

—

We therefore amended the manuscript on page 9 and added the following sentences: “Here, we identified 109 unique lysine-lysine contact sites (uxIDs) within full-length E6AP in the absence of E6, corresponding to 379 unique crosslinks (containing 204 light-light, 168 heavy-heavy and 7 light-heavy links) that were identified in total over 3 biological replicates (Supplementary Tables 9). In the presence of E6 we identified 108 unique lysine-lysine contact sites (uxIDs) within full-length E6AP, corresponding to 383 unique crosslinks (containing 204 light-light, 172 heavy-heavy and 7 light-heavy links) (Supplementary Table 10).”

And on page 15: “ We also noticed that the number of uxIDs are lower in experiments, where SILAC was used in combination with crosslinking. This may be explained by a direct effect of SILAC on the efficiency of crosslink identification. However, as the number of overall identified unique crosslinks are roughly similar in SILAC and non-SILAC crosslinking experiments, this effect may alternatively be explained by the relative smaller amount of peptides that was loaded on the column (i.e. the same total peptide amount loaded onto the LC MS/MS column should translate to half the amount of light E6AP peptides being present in the SILAC vs the non-SILAC sample).

7. “However, it is interesting to note that none of the detected interlinks is located within the C-terminal half of E6AP comprising the HECT domain, where the oligomerization site has been previously proposed to be located. In summary, our SILAC-XL-MS data suggests that E6AP can engage in homo-oligomeric interactions, but these are not mediated through the HECT domain,

and that oligomerization is not initiated or intensified by E6." No crosslinker detected at the HECT domain does not necessarily mean no binding in this region. This sentence should be rephrased.

We agree with the reviewer that the absence of detected crosslink does not necessarily mean that there is no interaction in this region. This is also the reason why we tried to carefully phrase our finding by stating that our crosslinking data only **suggest** that oligomerization is not mediated via the HECT domain of E6AP.

In order to make this statement even clearer, we changed the following sentences on page 10: "In summary, **while the absence of crosslinks does not unambiguously prove that there is no interaction**, our SILAC-XL-MS data suggests that E6AP can engage in homo-oligomeric interactions, but these are **likely** not mediated through the HECT domain, and that oligomerization is not initiated or intensified by E6."

8. Lastly, I would like to give a general comment of the supplementary tables. Merely listing all cross-links in different experiments is tedious, and very difficult for the readers to seek for useful information. I suggest the authors change the way of presenting crosslinks to format that is similar to typical quantitative proteomics experiments, listing all identified crosslinks, their normalized intensities (indicating missing values), across all biological replicates and conditions in comparison, as well as t-test significance and p-value. In this way, it is straightforward for the readers to immediately see the quality of the data and what cutoff the authors are chosen to report the significance.

We agree with the reviewer that the current structure of the supplementary tables is not optimal and it may indeed be complicated to find the information one is looking for - even though this was exactly what we had in mind when creating the current supplementary tables (one may even argue that it is also not always straightforward to find the relevant information in a typical quantitative proteomics experiment either).

An additional complication is given by the fact that our data processing pipeline is currently set-up in a way that the information containing the bulk part on the quantification (e.g. separate quantification of each peptide over different charge stages, fractions, multiple peptides for the same unique crosslinking site, across replicates and conditions) is summed up at a stage prior to the final output files, and while a t-test is performed and significance is recorded (p-value, FDR, Bonf), the test statistic is not recorded yet. The reviewer's comment is a good incentive for us to intensify our efforts to optimize this part of the workflow. While it is beyond the scope of this manuscript to code and implement those changes, we are more than happy to make these additional files available to the reviewer, if he or she wishes so.

We also deliberately had decided against uploading the complete output tables, as in our experience many readers, in particular the ones not so familiar with mass spectrometry, are side-tracked from the relevant information by the many additional columns, particularly the MS specific ones.

However, raw data, complete xQuest and xProphet (including information on charge states, retention times, p-values, FDRs, etc) have been deposited in PRIDE (px-submission #277951; Submission Reference: 1-20180530-44244).

Reviewers' comments:

Reviewer #2 (Remarks to the Author):

I would like to thank the authors for the comprehensive response, resolving several doubts, uploading the models and data, performing additional modeling runs and acknowledging and fixing the little mistake in the modeling runs.

Still, the modeling part requires clarifications. It still leaves open questions and doubts. The main point is that it is not clear whether 1) the modeling is driven by too few restraints or 2) the sufficient number of restraints is used but this is just not clearly presented.

I would suggest either improving the quality of the presentation of the modeling part (see points below) or moving the modeling part to the supplementary data and notes in the discussion, especially that authors mention that "our modeling framework is merely used as an additional tool to validate our findings, which in principle we could also omit without losing critical information."

Point 1:

Regarding my comment:

- "The E6AP – E6 modeling is based on only 4 crosslinks, and E6AP – E6 – p53 modeling based on 7 crosslinks. It is not much in absence of complementary data. How would the results change if some of the crosslinks are excluded? How would the results change if you use Id-score threshold of 30? "

And the response:

"The E6AP-E6 modeling was actually based on 13 interlink crosslinks for the E6AP-E6 interface, and 19 interlinks for the E6AP-E6-p53 interface since unique crosslinked peptides (e.g. multiple unique peptides can link one unique crosslinking site) were used as input as modeling restraints."

By "only 4/7 crosslinks" of course I meant 4/7 unique residue pairs = 4/7 uxIDs = 4/7 modeling restraints, I apologize for not being clear. Do you mean you used 13 uxIDs or 13 crosslinks leading to four uxIDs?

If I understand correctly, from the figure 4c (the model of E6AP – E6) one can see that there were only four uxIDs used for modeling (even if supported by 13 crosslinks leading to the same lysine-lysine pairs). Out of the four uxIDs (red lines), three of them group together and one locates separately from the rest (indicated by an arrow and comment on the figure attached to this review). Thus, the orientation between the proteins is heavily influenced by this specific crosslink. What if this crosslink is a false positive? Would the model be affected if it is excluded from modeling? It does look like it could change significantly as this part would not be restrained at all.

Regarding the response: "As suggested by the reviewer, we have tested how using a crosslinking dataset, where 15 percent of the links were randomly removed (jackknifing), as well as limiting the dataset to links with an Id-score >30, would influence our modeling results."

Was the jackknifing performed on the set of 13 crosslinks or the four uxIDs? It should be done at the uxID level, because 15% jackknifing of 13 redundant crosslinks can lead to the four uxIDs in most of the jackknifing samples, not changing anything in most of the jackknifing runs, as they will be run with the complete set of restraints. With four uxIDs only, perhaps you could run four modeling runs, each time removing one of the four restraints, and check whether the model from any of the any of the four runs would lead to different conclusions?

Likewise, it is unclear how many uxIDs were remaining after Id-score >30 filtering and which of the four (or seven) crosslinks, if any, were excluded from modeling at this threshold.

B

E6AP	E6
7	65
7	72
779	72
799	72
350	94
7	108
841	108
708	122

Which of these uxIDs were used for modeling?

Are the four highlighted the ones used for modeling?

E6AP 1-60 AZUL domain 150-200 HERC2 binding 378-395 E6 binding 495-852 HECT domain
 E6 30-66 zinc finger 103-139 zinc finger 147-151 PDZ domain
 Ubch7 4-147

Point 2:

Regarding the discussion:

Reviewer Comment: "The complex is small and could be addressed by high-resolution modeling/docking such as Haddock or Rosetta instead of the coarse-grained modeling used. This would add a possibility of using physical potential and scoring to aid modeling. One could argue that the other methods do not utilize Bayesian crosslinking restraint, but the benefit of this restraint is not clear in this work. Could the authors try to compare their model to a model obtained using Haddock webserver, which should be straightforward to run for this case?"

Authors Response: "The reviewer is correct in pointing out that using physical potential and scoring may potentially help the modeling. However, since we only included regions with a known crystal structure in the final modeling runs, we decided that using the crosslinks as (bayesian) distance restraints in conjunction with the excluded volume restraint should suffice in our case. However, if the reviewer feels that additional high-resolution modeling/docking by Haddock/Rosetta is still needed to support our modeling results, we will carry those out."

I do not understand this reasoning. Having crystal structures exactly enables you using higher resolution physical potential instead of the lower resolution excluded volume restraint. Especially for small computationally tractable systems. If the modeling is indeed performed based on four uxIDs for E6AP-E6 and seven for E6AP-E6-p53, as it still looks to me, you could actually benefit from using additional restraints of the physical potential. I leave the decision to the authors whether to follow this suggestion, but I would really encourage trying to run at least Haddock webserver. This comment may become obsolete if my doubts regarding point 1 can be addressed

(and indeed the higher number of restraints is used).

Reviewer #3 (Remarks to the Author):

1. In part 1 of the result “Crosslinking of E6AP in the absence and presence of the E6 oncoprotein reveals distinctive crosslink patterns”, the authors show possible conformation change of E6AP upon binding to E6, as illustrated by diminished crosslinks within the central region of E6AP. By merely visualizing the crosslink pattern shown in Figure 1A, B and C, I could see two crosslinks are missing within the central region of E6AP when binding to E6. However, in part 2 of the results, “SILAC-XL-MS reveals weak interactions between N-terminal regions of E6AP, but not the HECT domain”, these two significantly changed crosslinks did not appear in Figure 2B. I wonder if they were not identified in this dataset or was not able to quantify due to missing values. I think the authors should give reasons for this issue explicitly. We presume that the reviewer refers to the following links: 327:x:529; 300:x:779 (same holds true for 529:x:708; 327:x:412). As the reviewer correctly points out, those links are only shown in Figure 1B but not in the quantitative dataset in Figure 2B (we assume that the reviewer is referring to “binding of E6 induces a conformational rearrangement of E6AP bringing N- and C-termini into closer proximity” and the list of significantly up or down regulated crosslinks in Figure 2 rather than to the list of links between E6AP protomers in “SILACXL-MS reveals weak interactions between N-terminal regions of E6AP, but not the HECT domain”). The reason is that those links were identified but not quantified according to our cut-off criteria. While only links that were identified with high confidence (e.g. Id-Score of >25, deltaS Score 0 or a p-value of $\geq \pm 0.01$) will not show up as a quantified link in Figure 2 (i.e. not even as a grey line). Only if the link additionally has undergone a fold change of $\log_2\text{ratio} \geq \pm 1.5$ between conditions will the link be considered significantly up- or down-regulated and is shown in green or red, respectively. In the particular case of the links mentioned above, the reason for their non-appearance in Figure 2 has been that they had violations >0; i.e.: 327:x:529 (2 violations); 300:x:779 (2 violations); 529:x:708 (1 violation); 327:x:412(2 violations).

I still find part 1 and part 2 is redundant and confusing, therefore I prefer they are merged. I understand that some identified crosslinks cannot be quantified due to various reasons. However, if I understand it correctly, the author concludes that the two crosslinks (327:x:529; 300:x:779) show significant changes using only identification while not quantifiable due to violation when using quantitative information. Shall I believe these cross-links are changed or not? The authors argue that part 1 gives global and nonquantitative information in their rebuttal. In this case, one shouldn't make any quantitative/comparative conclusions. Therefore, I think “Remarkably, in addition to the expected interlinks between E6AP and E6, the presence of E6 had a noticeable effect on the crosslinking pattern within E6AP itself, leading to diminished crosslinking within the central region of the protein (i.e. the region of E6AP between the AZUL and the HECT domain) (Fig. 1C)” as well as other comparative conclusions in the same paragraph should be removed.

2. Regarding supplementary Tables, the authors argue that “We also deliberately had decided against uploading the complete output tables, as in our experience many readers, in particular the

ones not so familiar with mass spectrometry, are sidetracked from the relevant information by the many additional columns, particularly the MS specific ones.”. Without quantitative information, a list of identified crosslinks is useless to both MS and non-MS scientists. As an alternative, the authors can upload a cleaned-up xProphet output as a Supplementary Dataset in excel format.

Point-by-point responses to reviewers' comments on "Structural dynamics of the E6AP/UBE3A-E6-p53 enzyme-substrate complex" (NCOMMS-18-11592A)

Reviewer #2 (Remarks to the Author):

I would like to thank the authors for the comprehensive response, resolving several doubts, uploading the models and data, performing additional modeling runs and acknowledging and fixing the little mistake in the modeling runs.

Still, the modeling part requires clarifications. It still leaves open questions and doubts. The main point is that it is not clear whether 1) the modeling is driven by too few restraints or 2) the sufficient number of restraints is used but this is just not clearly presented. I would suggest either improving the quality of the presentation of the modeling part (see points below) or moving the modeling part to the supplementary data and notes in the discussion, especially that authors mention that "our modeling framework is merely used as an additional tool to validate our findings, which in principle we could also omit without losing critical information."

Point 1: *Regarding my comment: - "The E6AP – E6 modeling is based on only 4 crosslinks, and E6AP – E6 – p53 modeling based on 7 crosslinks. It is not much in absence of complementary data. How would the results change if some of the crosslinks are excluded? How would the results change if you use Id-score threshold of 30? "And the response: "The E6AP-E6 modeling was actually based on 13 interlink crosslinks for the E6AP-E6 interface, and 19 interlinks for the E6AP-E6-p53 interface since unique crosslinked peptides (e.g. multiple unique peptides can link one unique crosslinking site) were used as input as modeling restraints."*

By "only 4/7 crosslinks" of course I meant 4/7 unique residue pairs = 4/7 uxIDs = 4/7 modeling restraints, I apologize for not being clear. Do you mean you used 13 uxIDs or 13 crosslinks leading to four uxIDs? If I understand correctly, from the figure 4c (the model of E6AP – E6) one can see that there were only four uxIDs used for modeling (even if supported by 13 crosslinks leading to the same lysine-lysine pairs). Out of the four uxIDs (red lines), three of them group together and one locates separately from the rest (indicated by an arrow and comment on the figure attached to this review). Thus, the orientation between the proteins is heavily influenced by this specific crosslink. What if this crosslink is a false positive? Would the model be affected if it is excluded from modeling? It does look like it could change significantly as this part would not be restrained at all. Regarding the response: "As suggested by the reviewer, we have tested how using a crosslinking dataset, where 15 percent of the links were randomly removed (jackknifing), as well as limiting the dataset to links with an Id-score >30, would influence our modeling results." Was the jackknifing performed on the set of 13 crosslinks or the four uxIDs? It should be done at the uxID level, because 15% jackknifing of 13 redundant crosslinks can lead to the four uxIDs in most of the jackknifing samples, not changing anything in most of the jackknifing runs, as they will be run with the complete set of restraints. With four uxIDs only, perhaps you could run four modeling runs, each time removing one of the four restraints, and check whether the model from any of the any of the four runs would lead to different conclusions? Likewise, it is unclear how many uxIDs were remaining after Id-score >30 filtering and which of the four (or seven) crosslinks, if any, were excluded from modeling at this threshold.

We thank the reviewer for their appreciation of our "comprehensive response (and....) additional modeling runs".

The reviewer is correct in their assumption that we were indeed using 13 unique crosslinks leading to four uxIDs in the case of our model of the E6AP-E6 interface. Of these uxIDs indeed "three of them group together and one locates separately from the rest" and we understand that the reviewer is particularly concerned if "the orientation between the proteins may be influenced by this specific crosslink".

Following the reviewer's suggestion, we have therefore excluded **all** crosslinks (i.e. two unique crosslinks) from the *uxID* E6AP_K708 – E6_K122 (i.e. the crosslink highlighted by the reviewer with a red arrow).

Figure 1 shows the original structural model of the binding interface between the HECT domain of E6AP and E6 overlaid with the density map from the modeling run missing all crosslinks between E6AP_K708 – E6_K122 (violet). This clearly shows that the exclusion of these crosslinks results in a marginal shift in the orientation of the density map of E6 in regard to the one of E6AP. The run contains altogether five roughly equally populated clusters that are all centred around the same position and where the general orientation of the density maps remains the same, with only slightly different shifts in their respective angles. Excluding the restraints from the E6AP_K708 – E6_K122 *uxID* does therefore not severely affect the model.

Figure 1: Structural model of the binding interface between the HECT domain of E6AP and E6. Shown is the original structural alignment of the proteins using the center model of E6 overlaid with the density map from the modeling run missing all crosslinks between E6AP_K708 – E6_K122 (violet).

We then proceeded as suggested by the reviewer and removed in iterative runs always one of the 3 remaining *uxID* restraints – always removing all unique crosslinks for a particular *uxID* (**Figure 2**).

Figure 2: Structural models of the binding interface between the HECT domain of E6AP and E6. Shown are the original structural alignment of the proteins including the center model for E6 overlaid with the density maps from the various modeling runs missing all crosslinks between E6AP_K779 – E6_K72 (**A**), E6AP_K799 – E6_K72 (**B**) or E6AP_K841 – E6_K108(**C**), respectively.

As can be clearly seen from the models, the complete exclusion of two additional *uxID* restraints (E6AP_K779 – E6_K72 and E6AP_K799 – E6_K72) has virtually no influence on our models, whereas in once case (E6AP_K841 – E6_K108) removal of all unique crosslinks leads to a discernible shift in the respective density map. However, even in this model the main findings from our study – i.e. the binding site of E6 is in the vicinity of the catalytic centre of E6AP and distinct from the binding site of Ubch7, the cognate E2 of E6AP - is still valid.

We would also like to mention that we consider it to be highly unlikely that **all** detected links *for a uxID* are false-positives, as we have not only carried out extensive validation of all of our crosslinks but have also identified crosslinks from multiple, completely independent biological replicates for each of the 4

uxIDs. Moreover, we also find the vast majority of these links again using a completely different particle (i.e. the one containing also p53).

In order to hopefully eliminate any potentially remaining last doubts, we additionally used our crosslinking restraints as input for HADDOCK. **Figure 3** shows the structural model of the binding interface between the HECT domain of E6AP and E6 generated by HADDOCK which we superimposed on the density map from the original structural model from our IMP based modeling runs by aligning the E6AP HECT domains from the HADDOCK and IMP models. As can be seen, the model using HADDOCK generates a nearly identical binding interface between the HECT domain of E6AP and E6, where the E6 structure of the HADDOCK model fits almost completely into the E6 density of the IMP model.

Figure 3: Structural model of the binding interface between the HECT domain of E6AP and E6 using HADDOCK, which was superimposed to the density map from the original structural model from our IMP based modeling runs by aligning the E6AP HECT domains from the HADDOCK and IMP models. Structural calculation of the complex with HADDOCK was done using a combination of crosslink distance restraints and interface information derived from DisVis analysis. The DisVis webserver (<http://milou.science.uu.nl/cgi/services/DISVIS/disvis/submit>) allows to perform an interaction analysis which defines surface residues on E6AP and E6 that are putatively involved in the interaction interface of the two molecules. Here, E6AP was chosen as the fixed chain, E6 as the scanning chain and a restraints file was generated from the experimentally determined crosslinks. Additionally, amino acid residues of E6AP and E6 with > 40% surface accessibility were determined with a web tool (<http://cib.cf.ocha.ac.jp/bitool/ASA/>) and specified during DisVis analysis. Residues with an average number of interactions > 0.5 were defined as active residues during HADDOCK docking (<http://haddock.science.uu.nl/services/HADDOCK2.2/haddockserver-expert.html>). Passive residues were automatically defined within a radius of 6.5 Å around the active residues. The experimentally defined crosslinks were used to supply HADDOCK with a TBL file containing unambiguous restraints. The default number of models has been generated (1000 for rigid body docking and 200 for semi-flexible and water-refinement) and the final models were automatically clustered based on the fraction of common contacts (FCC) which measures the similarity of intermolecular contacts. In summary, HADDOCK clustered 166 structures in 12 clusters which represent 83 % of the water-refined models (total of 200 by default). Cluster 1 - which is shown in **Figure 3** - is the top cluster with a Z-Score of -1.5. It contains 33 structures.

In summary, both our additional modeling runs, where we performed a rigorous jackknifing approach, in which complete *uxIDs* were purposely removed, as well as an independent modeling approach using HADDOCK clearly confirm that our modeling results are not driven by too few restraints but are highly robust and reproducible. Nevertheless, in order to indicate that our modeling results are not entirely unambiguous under the assumption that large parts of the data are false-positives and are therefore purposely removed, we have added the new modeling runs from **Figure 1-3** into a new **Supplementary Figure 8** and have added the following sentence to the manuscript on page 24: “Additional modeling runs, in which all interlinks between E6AP and E6 for a specific *uxID* were purposely removed, as well as an independent modeling approach using HADDOCK confirm the overall robustness and reproducibility of our modeling results(Supplementary Fig. 8).”

Point 2: Regarding the discussion: Reviewer Comment: “The complex is small and could be addressed by high-resolution modeling/docking such as Haddock or Rosetta instead of the coarse-grained

modeling used. This would add a possibility of using physical potential and scoring to aid modeling. One could argue that the other methods do not utilize Bayesian crosslinking restraint, but the benefit of this restraint is not clear in this work. Could the authors try to compare their model to a model obtained using Haddock webserver, which should be straightforward to run for this case?” Authors Response: “The reviewer is correct in pointing out that using physical potential and scoring may potentially help the modeling. However, since we only included regions with a known crystal structure in the final modeling runs, we decided that using the crosslinks as (bayesian) distance restraints in conjunction with the excluded volume restraint should suffice in our case. However, if the reviewer feels that additional high-resolution modeling/docking by Haddock/Rosetta is still needed to support our modeling results, we will carry those out.” I do not understand this reasoning. Having crystal structures exactly enables you using higher resolution physical potential instead of the lower resolution excluded volume restraint. Especially for small computationally tractable systems. If the modeling is indeed performed based on four uxIDs for E6AP-E6 and seven for E6AP-E6-p53, as it still looks to me, you could actually benefit from using additional restraints of the physical potential. I leave the decision to the authors whether to follow this suggestion, but I would really encourage trying to run at least Haddock webserver. This comment may become obsolete if my doubts regarding point 1 can be addressed (and indeed the higher number of restraints is used).

As discussed above, using HADDOCK leads to a virtually identical binding interface between the HECT domain of E6AP and E6, thus strongly corroborating our previous models generated using IMP.

Reviewer #3 (Remarks to the Author):

1. In part 1 of the result “Crosslinking of E6AP in the absence and presence of the E6 oncoprotein reveals distinctive crosslink patterns”, the authors show possible conformation change of E6AP upon binding to E6, as illustrated by diminished crosslinks within the central region of E6AP. By merely visualizing the crosslink pattern shown in Figure 1A, B and C, I could see two crosslinks are missing within the central region of E6AP when binding to E6. However, in part 2 of the results, “SILAC-XL-MS reveals weak interactions between N-terminal regions of E6AP, but not the HECT domain”, these two significantly changed crosslinks did not appear in Figure 2B. I wonder if they were not identified in this dataset or was not able to quantify due to missing values. I think the authors should give reasons for this issue explicitly.

We presume that the reviewer refers to the following links: 327:x:529; 300:x:779 (same holds true for 529:x:708; 327:x:412). As the reviewer correctly points out, those links are only shown in Figure 1B but not in the quantitative dataset in Figure 2B (we assume that the reviewer is referring to “binding of E6 induces a conformational rearrangement of E6AP bringing N- and C-termini into closer proximity” and the list of significantly up or down regulated crosslinks in Figure 2 rather than to the list of links between E6AP protomers in “SILACXL- MS reveals weak interactions between N-terminal regions of E6AP, but not the HECT domain”). The reason is that those links were identified but not quantified according to our cut-off criteria. While only links that were identified with high confidence (e.g. Id-Score of >25, deltaS Score <0.95 and identification in at least two out of 3 independent biological replicates) were admitted to our quantitation pipeline, only a subset of those links also passed our cut-off criteria for quantification (violations = 0; log2ratio $\geq \pm 1.5$ and a p-value of ≤ 0.01). At this point, it is also important to emphasize that while identification of crosslinks is happening on the level of the unique peptide (i.e. unique peptide sequence and linkage site), quantification is done on the level of the unique crosslinking site (uxID), i.e. multiple different crosslinked peptides can contribute to a unique crosslinking site. Therefore, a crosslink that has a violation of >0 or a p-value of $\geq \pm 0.01$ will not show up as a quantified link in Figure 2 (i.e. not even as a grey line). Only if the link additionally has undergone a fold change of log2ratio $\geq \pm 1.5$ between conditions will the link be considered significantly up- or down-regulated and is shown in green or red, respectively. In the particular case of the links mentioned above, the reason for their non-appearance in Figure 2 has been that they had violations >0; i.e.: 327:x:529 (2 violations); 300:x:779 (2 violations); 529:x:708 (1 violation); 327:x:412 (2 violations).

I still find part 1 and part 2 is redundant and confusing, therefore I prefer they are merged.

As requested, we have now merged parts 1 and 2. Also Figure 1 and 2 have therefore been combined into one new Figure 1.

I understand that some identified crosslinks cannot be quantified due to various reasons. However, if I understand it correctly, the author concludes that the two crosslinks (327:x:529; 300:x:779) show significant changes using only identification while not quantifiable due to violation when using quantitative information. Shall I believe these cross-links are changed or not? The authors argue that part 1 gives global and nonquantitative information in their rebuttal. In this case, one shouldn't make any quantitative/comparative conclusions. Therefore, I think "Remarkably, in addition to the expected interlinks between E6AP and E6, the presence of E6 had a noticeable effect on the crosslinking pattern within E6AP itself, leading to diminished crosslinking within the central region of the protein (i.e. the region of E6AP between the AZUL and the HECT domain) (Fig. 1C)" as well as other comparative conclusions in the same paragraph should be removed.

We apologize for not properly explaining why we concluded 'that the two crosslinks (327:x:529; 300:x:779) show significant changes using only identification while not (being) quantifiable due to violation' in the first round of revision.

As stated before, for identification a crosslink was shown in Figure 1 if it was identified in 2 of 3 biological replicates – in any of the 4 SEC fractions which have been measured in technical duplicates on the mass spectrometer. In contrast, for quantification, violations were calculated for each **charge** state of a *uxID*. We included this additional layer of scrutiny because quantitative XL-MS is still relatively novel and we wanted to make sure that only links that we were able to quantify in a highly consistent manner were included in our analysis.

This is why a given crosslink may be included in former Figure 1 but not in former Figure 2 (now both combined in the new Figure 1, see above). In order to clarify this point, we have now also included those links in the new combined Figure 1 (with a dashed grey line) that were identified but did not stand our additional level of scrutiny used for quantitation.

For clarification purposes, we have now added *uxIDs* to Supplementary Tables 5 & 6 with violations > 0. Thus, also the previously discussed linking sites (327:x:529; 300:x:779; 529:x:708; 327:x:412) are now included in supplementary Table 5 with a log2 ratio. Since these links were only reliably identified in the crosslinking experiments without E6 but not in the experiments with E6 (and the experiments without E6 are set as the reference experiments), we would expect a negative log2 ratio for these *uxIDs* (Table 1). This is clearly the case for all discussed links – so the answer is yes, one should indeed believe that those links are changed.

crosslinking site	log2 ratio	pvalue	sum violations
327:x:529	-7.69	0.00	2
300:x:779	-3.52	0.00	2
529:x:708	-3.06	0.00	1
327:x:412	-6.12	0.00	2

Table 1

We hope this clarifies that there are no inconsistencies between qualitative and quantitative crosslinking data.

2. Regarding supplementary Tables, the authors argue that "We also deliberately had decided against uploading the complete output tables, as in our experience many readers, in particular the ones not so familiar with mass spectrometry, are sidetracked from the relevant information by the many additional columns, particularly the MS specific ones.". Without quantitative information, a list of identified crosslinks is useless to both MS and non-MS scientists. As an alternative, the authors can upload a cleaned-up xProphet output as a Supplementary Dataset in excel format.

We have now added supplementary data that provide detailed quantitative information for every peptide identified (additional Supplementary Tables 13/14 for E6AP +/- E6 and 15/16 for E6AP +/- E6_L50E, respectively). With these additional files it is now possible to get detailed quantitative information for every peptide (for each charge state, type (e.g. xlink:heavy, xlink:light) over states and experiments and follow their integration into the final calculation of the fold change (log2 ratios)).

As described previously, *xTract* calculates fold changes (log₂ ratios) of *uxIDs*. A *uxID* is defined by the positions of the crosslinked lysine residues in the respective proteins (e.g. E6AP:860:x: E6AP:97). For every *uxID*, multiple *peptideIDs* (peptide sequences) can be found in one experiment (different tryptic peptides for the same linking site). Every *peptideID* can be found in multiple charge states which is called a *uID* (peptide sequence, the charge state and the type (e.g. xlink:heavy, xlink:light)).

In Supplementary Table 1-3, the crosslinked peptides (*peptideIDs*) are listed. Based on the *xTract* output tables, it is possible to calculate intensities for the *peptideIDs*. However, the *xTract* software applies some validation steps on the *uID* level. For example, the 'compare labels' validation: As we are using a heavy and a light labeled DSS crosslinker in our experiments, for every technical replicate of all biological replicates the sums and the max values are used to calculate the mean of the sums or the mean of the max values for each label. These values are compared and a violation is recorded, if the differences are larger than a threshold value. Thus, not all *uIDs* belonging to one *peptideID* are considered in order to calculate the log₂ ratio. Because of these additional validation steps, our experiments differ from typical quantitative proteomics experiments. In our opinion, it is therefore better to show additional *bagcontainer* output tables of *xTract* in the supplementary data than to calculate intensities for the *peptideIDs* and include those in the existing supplementary tables.

We therefore included the *xTract* output files *bagcontainer.stats* and *bagcontainer.details.stats* as supplementary tables 13 and 14 (E6AP +/- E6) and 15 and 16 (E6AP +/- E6_L50E). The *bagcontainer.details.stats* table contains the summed areas of all isotopes belonging to one peakgroup. The areas are indicated for every *uID* in all measurements separately. The *bagcontainer.stats* table contains the *msum_area_sum_isotopes* in condition T1 (E6AP only) and T2 (E6AP +/- E6) for every *uID*. The log₂ ratio for a *uxID* is calculated from the ratio T2/T1 of the summed *msums* for heavy and light of all *uIDs* referring to one *uxID*. By default, only *uIDs* with violation=0 in T1 and T2 are considered for log₂ calculation. The log₂ ratio for a specific *uxID* can be found in supplementary table 5 and 6 (as before).

Additionally, we have also uploaded all raw and *xProphet* output files for every experiment on PRIDE (Project accession: PXD010002).